# LogSpecT: Feasible Graph Learning Model from Stationary Signals with Recovery Guarantees

**Shangyuan Liu**
The Chinese University of Hong Kong
`shangyuanliu@link.cuhk.edu.hk`

**Linglingzhi Zhu**
The Chinese University of Hong Kong
`llzzhu@se.cuhk.edu.hk`

**Anthony Man-Cho So**
The Chinese University of Hong Kong
`manchoso@se.cuhk.edu.hk`

## Abstract

Graph learning from signals is a core task in graph signal processing (GSP). A significant subclass of graph signals called the stationary graph signals that broadens the concept of stationarity of data defined on regular domains to signals on graphs is gaining increasing popularity in the GSP community [10, 20, 25]. The most commonly used model to learn graphs from these stationary signals is *SpecT* [31], which forms the foundation for nearly all the subsequent, more advanced models. Despite its strengths, the practical formulation of the model, known as *rSpecT*, has been identified to be susceptible to the choice of hyperparameters. More critically, it may suffer from infeasibility as an optimization problem. In this paper, we introduce the first condition that ensures the infeasibility of rSpecT and design a novel model called *LogSpecT*, along with its practical formulation *rLogSpecT* to overcome this issue. Contrary to rSpecT, our novel practical model rLogSpecT is always feasible. Furthermore, we provide recovery guarantees of rLogSpecT from modern optimization tools related to epi-convergence, which could be of independent interest and significant for various learning problems. To demonstrate the practical advantages of rLogSpecT, a highly efficient algorithm based on the linearized alternating direction method of multipliers (L-ADMM) that allows closed-form solutions for each subproblem is proposed with convergence guarantees. Extensive numerical results on both synthetic and real networks not only corroborate the stability of our proposed methods, but also highlight their comparable and even superior performance than existing models.

## 1 Introduction

Learning with graphs has proved its relevance in many practical areas, such as life science [36, 37], signal processing [13, 14, 38, 39], and financial engineering [1, 21], to just name a few. However, there are many cases that the graphs are not readily prepared and only the data closely related to the graphs can be observed. Hence, a core task in Graph Signal Processing (GSP) is to learn the underlying graph topology based on the interplay between data and graphs [22].

Among these commonly used data properties, an assumption that is referred to as the graph signal stationarity [10] is gaining attention [30, 20, 25]. Graph signal stationarity is the extension of the notion of stationarity from the regular space/time domain to the irregular domain (i.e. graphs). The latter is a traditional hypothesis in signal processing to capture a special type of statistical relationship between samples of a temporal signal. It is a cornerstone of many signal processing methods. Although the graph signal stationarity property is proposed more from a theoretic end, the

37th Conference on Neural Information Processing Systems (NeurIPS 2023).

techniques based on this idealistic assumption pave the way for the development of the following more advanced and applicable models [35, 34]. Furthermore, several works have shown that some real datasets can be approximately viewed as stationary or partially explained by the stationarity assumption. For instance, [25] revealed that the well-known USPS dataset and the CMUPIE set of cropped faces exhibit near stationarity. [10] found that certain weather data could be explained by stationary graph signals. Additionally, [31] highlighted the relevance of the stationarity assumption in learning protein structure. [43] applied the stationary assumption to US Senate Roll Calls and achieved good performance.

The predominant methods to process stationary graph signals and learn topology under the stationarity assumption are the spectral-template-based models. The start of this line of works is [31], which proposed a vanilla model called *SpecT* [31] to learn graphs from stationary graph signals. Many extensions of this fundamental model have been made since then to accommodate the more realistic and complicated settings, e.g. joint inference of multiple graphs [32, 27], learning with streaming graph signals [33], learning with hidden nodes [4], learning from both smooth and stationary graph signals [5]. In practice, SpecT requires the unknown data covariance matrix. Hence, a robust formulation called *rSpecT* is proposed [31], which replaces the unknown covariance matrix with its estimate and introduces a hyperparameter to reflect the estimation inaccuracy. Unfortunately, the model is sensitive to this hyperparameter and improper tuning of it may jeopardize the model performance or lead to model infeasibility.

The current approach to selecting an appropriate value is heuristic. It finds the smallest one that allows for a feasible solution. This method has two shortcomings. On one hand, It is computationally costly. The smallest value is the exact solution to a second-order conic programming, which is solved approximately with computational methods. Any approximate solution that is smaller than the exact one will make rSpecT infeasible. On the other hand, such an approach lacks theoretical guarantees as how the smallest value behaves as the number of observations increases is unknown. This issue makes the existing recovery guarantees invalid. To the best of our knowledge, only a few works have focused on how to understand the performance of the robust formulation rSpecT. [24] empirically showed that the feasible set of the robust formulation approximates the original one as more samples are collected. [23] provided the first recovery guarantee for rSpecT. Their work exploits the fact that the objective is the $\ell_1$-norm and the constraints are linear [44]. However, the conditions needed in their work are not only restrictive but also hard to check, as they require the full rankness of a large-scale matrix related to random samples. Another model [34] tries to circumvent the model infeasibility issue by turning a constraint into a penalty. However, this approach introduces another hyperparameter that is neither easy to tune nor amenable to theoretical analysis. Also, this hyperparameter should go to infinity heuristically when more and more samples are collected, which may exacerbate model instability.

In this paper, we propose an alternative formulation of SpecT that can bypass the aforementioned issues. We first provide a condition that ensures the infeasibility of the fundamental model rSpecT. Then, we propose a novel formulation called *LogSpecT* and its practical formulation *rLogSpecT* based on the spectral template and incorporating a log barrier. The inclusion of a log barrier as a regularizer on the degrees is inspired by its extensive application in graph learning from smooth graph signals [15, 16, 19, 47] and the discussion in [7] emphasizing that restricting the degrees to preclude isolated nodes is crucial to learning graphs with special structures. It is worth noting that the approach in [7], which directly imposes constraints, may exacerbate the infeasibility issue in our problem. Although rLogSpecT still requires a parameter to reflect the estimation error of the unknown covariance matrix, it no longer suffers from the infeasibility issue associated with the improper choice of the parameter. Furthermore, we provide the decaying rate of the parameter that can guarantee the recovery of rLogSpecT. This not only aids in the parameter tuning, but also establishes a connection between rLogSpecT and LogSpecT through non-asymptotic recovery analysis. These theoretical results are derived using contemporary optimization tools related to epi-convergence [29]. Unlike the current guarantees for rSpecT that are built on an $\ell_1$ analysis model [44], our approach based on epi-convergence is not confined to the specific types of optimization problems (e.g. the combination of the $\ell_1$-norm and log barrier in the objective function) and consequently admits broader applications that can be of independent interest.

In the algorithmic aspect, we design a linearized alternating direction method of multipliers (L-ADMM) to solve rLogSpecT. The subproblems of L-ADMM admit closed-form solutions and can be implemented efficiently due to the linearization technique. Also, we provide the convergence result of the proposed method. Finally, we conduct extensive experiments on both synthetic data and

real networks. Notably, the infeasibility issue of rSpecT frequently occurs in experiments on real networks, thereby emphasizing its practical significance beyond theoretical concerns. Furthermore, we observe that even when the classical models are set to be feasible, our novel models (LogSpecT and rLogSpecT) exhibit improved accuracy and stability, particularly in the case of synthetic BA graphs. This empirically illustrates that although our models are initially proposed to circumvent the infeasibility issue, they are superior alternatives to classical methods in the general cases.

## 1.1 Notation

The notation we use in this paper is standard. We use $[m]$ to denote the set $\{1, 2, \ldots, m\}$ for any positive integer $m$. Let the Euclidean space of all real matrices be equipped with the inner product $\langle X, Y \rangle \coloneqq \operatorname{tr}(X^\top Y)$ for any matrices $X, Y$ and denote the induced Frobenius norm by $\|\cdot\|_F$ (or $\|\cdot\|_2$ when the argument is a vector). For any $X \in \mathbb{R}^{m \times n}$, we use $\mathbb{B}(X, \rho)$ to denote the closed Euclidean ball centering at $X$ with radius $\rho$. Let $\|\cdot\|$ be the operator norm, $\|X\|_{1,1} \coloneqq \sum_{i,j} |X_{ij}|$, $\|X\|_{\infty,\infty} \coloneqq \max_{i,j} |X_{ij}|$, and let $\operatorname{diag}(X)$ be vector formed by the diagonal entries of $X$. For a column vector $x$, let $\operatorname{Diag}(x)$ be the diagonal matrix whose diagonal elements are given by $x$. Given a closed and convex set $\mathcal{C}$, we use $\Pi_{\mathcal{C}}(w) \coloneqq \operatorname{argmin}_{v \in \mathcal{C}} \|v - w\|_2$ to denote the projection of the point $w$ onto the set $\mathcal{C}$. We use $\mathbf{1}$ (resp. $\mathbf{0}$) to denote an all-one vector (resp. all-zero vector) whose dimension will be clear from the context. For a set $\mathcal{S}$ and any real number $\alpha$, let $\alpha\mathcal{S} \coloneqq \{\alpha x \mid x \in \mathcal{S}\}$, $\mathcal{S}\mathbf{1} \coloneqq \{x\mathbf{1} \mid x \in \mathcal{S}\}$, and $\iota_{\mathcal{S}}(\cdot)$ be the indicator function of $\mathcal{S}$. For two non-empty and compact sets $\mathcal{X}$ and $\mathcal{Y}$, the distance between them is defined as $\operatorname{dist}(\mathcal{X}, \mathcal{Y}) \coloneqq \sup_{X \in \mathcal{X}} \inf_{Y \in \mathcal{Y}} \|X - Y\|_F$[1].

## 2 Preliminaries

Let $\mathcal{G} = (\mathcal{V}, \mathcal{E})$ be a graph, where $\mathcal{V} = [m]$ is the set of nodes and $\mathcal{E} \subset \mathcal{V} \times \mathcal{V}$ is the set of edges. Let $S$ be the weight matrix associated with the graph $\mathcal{G}$, where $S_{ij}$ represents the weight of the edge between nodes $i$ and $j$. In this paper, we consider undirected graphs without self-loops. The set of adjacency matrices for such graphs is $\mathcal{S} \coloneqq \{S \in \mathbb{R}^{m \times m} \mid S = S^\top, \operatorname{diag}(S) = \mathbf{0}, S \geq \mathbf{0}\}$. Suppose that the adjacency matrix $S$ admits the eigen-decomposition $S = U \Lambda U^\top$, where $\Lambda$ is a diagonal matrix and $U$ is an orthogonal matrix. A graph filter is a linear operator $h : \mathbb{R}^{m \times m} \to \mathbb{R}^{m \times m}$ defined as $h(S) = \sum_{i=0}^{p} h_i S^i = U \left(\sum_{i=0}^{p} h_i \Lambda^i\right) U^\top$, where $p > 0$ is the order of the graph filter and $\{h_i\}_{i=0}^{p}$ are the filter coefficients. According to the convention, we have $S^0 = I$.

A graph signal can be represented by a vector $x \in \mathbb{R}^m$, where the $i$-th element $x_i$ is the signal value associated with node $i$. A signal $x$ is said to be stationary if it is generated from

$$x = h(S)w, \tag{1}$$

where $w$ satisfies $\mathbb{E}[w] = \mathbf{0}$ and $\mathbb{E}[w^\top w] = I$. Simple calculations give that the covariance matrix of $x$, which is denoted by $C_\infty$, shares the same eigenvectors with $S$. Hence, we have the constraint

$$C_\infty S = S C_\infty. \tag{2}$$

Based on this, the following fundamental model SpecT is proposed to learn graphs from stationary signals without the knowledge of graph filters [31, 34]:

$$\min_{S} \|S\|_{1,1}, \text{ s.t. } C_\infty S = S C_\infty, S \in \mathcal{S} \cap \left\{S \in \mathbb{R}^{m \times m} : \sum_{j=1}^{m} S_{1j} = 1\right\}, \tag{SpecT}$$

where the constraint $\sum_{j=1}^{m} S_{1j} = 1$ is used to preclude the trivial optimal solution $S^* = \mathbf{0}$. When $C_\infty$ is unknown and only $n$ i.i.d samples of $\{x_i\}$ from (1) are available, the robust formulation, which is based on the estimate $C_n$ of $C_\infty$ and called *rSpecT*, is used:

$$\min_{S} \|S\|_{1,1}, \text{ s.t. } \|C_n S - S C_n\|_F \leq \delta, S \in \mathcal{S} \cap \left\{S \in \mathbb{R}^{m \times m} : \sum_{j=1}^{m} S_{1j} = 1\right\}. \tag{rSpecT}$$

For this robust formulation, the recovery guarantee is studied empirically in [24] and theoretically in [23] under some conditions that are restrictive and hard to check.

---

[1] It reduces to the classic point-to-set (resp. point-to-point) distance when $\mathcal{Y}$ (resp. $\mathcal{X}$ and $\mathcal{Y}$) is a singleton.

## 3 Infeasibility of rSpecT and Novel Models

Even though rSpecT has gained much popularity, there are few works that discuss the choice of the hyperparameter $\delta$ and how it affects the model feasibility. In this section, we present a condition under which rSpecT is always infeasible and then propose an alternative formulation. To motivate our results, let us consider the following 2-node example.

**Example 3.1.** *Consider a simple graph containing 2 nodes. Then, the set given by the second constraint of rSpecT is $\mathcal{S} \cap \{\boldsymbol{S} \in \mathbb{R}^{m \times m} : \sum_{j=1}^m \boldsymbol{S}_{1j} = 1\} = [0, 1; 1, 0]$, which is a singleton. Suppose that the sample covariance matrix is $\boldsymbol{C}_n = [h_{11}, h_{12}; h_{12}, h_{22}]$. Then, the constraint $\|\boldsymbol{C}_n\boldsymbol{S} - \boldsymbol{S}\boldsymbol{C}_n\|_F \leq \delta$ is reduced to $2(h_{11} - h_{22})^2 \leq \delta^2$. Hence, when $h_{11} \neq h_{22}$ and $\delta < \sqrt{2}\,|h_{11} - h_{22}|$, rSpecT has no feasible solution.*

Before delving into the general case, we introduce a linear operator $\boldsymbol{B} \in \mathbb{R}^{m^2 \times m(m-1)/2}$ that maps the vector $\boldsymbol{x} \in \mathbb{R}_+^{(m-1)m/2}$ to the vectorization of an adjacency matrix $\boldsymbol{S} \in \mathcal{S}$ of a simple, undirected graph:

$$(\boldsymbol{Bx})_{m(i-1)+j} = \begin{cases} x_{i-j+\frac{i-1}{2}(2m-j)}, & \text{if } i > j, \\ 0, & \text{if } i = j, \\ x_{j-i+\frac{i-1}{2}(2m-i)}, & \text{if } i < j, \end{cases} \tag{3}$$

where $i, j \in [m]$. We also define $\boldsymbol{A}_n := \boldsymbol{I} \otimes \boldsymbol{C}_n - \boldsymbol{C}_n \otimes \boldsymbol{I} \in \mathbb{R}^{m^2 \times m^2}$, so that the first constraint of rSpecT can be rewritten as $\|\boldsymbol{C}_n\boldsymbol{S} - \boldsymbol{S}\boldsymbol{C}_n\|_F \leq \delta \iff \|\boldsymbol{A}_n \text{vec}(\boldsymbol{S})\|_2 \leq \delta$, where $\text{vec}(\cdot)$ is the vectorization operator. We now give a condition that guarantees the infeasibility of rSpecT.

**Theorem 3.2.** *Consider the linear system*

$$\boldsymbol{A}_n \boldsymbol{B} \boldsymbol{y} = \boldsymbol{0}, \ \ \boldsymbol{y} \leq \boldsymbol{0}, \ \ \sum_{i=1}^{m-1} y_i \neq \boldsymbol{0}, \tag{4}$$

*where $\boldsymbol{y} \in \mathbb{R}^{\frac{m(m-1)}{2}}$. If (4) has no feasible solution, then there exists a $\bar{\delta}_n > 0$ such that rSpecT is infeasible for all $\delta_n \in [0, \bar{\delta}_n)$.*

**Remark 3.3.** *From Theorem 3.2 we can infer that for any fixed $n$, the linear system (4) has no solution when $\boldsymbol{A}_n\boldsymbol{B}$ has full column rank (e.g., Example 3.1). This leads to the infeasibility of rSpecT with $\delta_n \in [0, \bar{\delta}_n)$. As remarked in [31], one should tackle the feasibility issue of rSpecT with caution.*

The failure of rSpecT (SpecT) lies in the existence of the constraint $(\boldsymbol{S1})_1 = 1$, which is used to preclude the trivial solution $\boldsymbol{S}^* = \boldsymbol{0}$. For this reason, we resort to an alternative approach to bypassing the trivial solution. When the graphs are assumed to have no isolated node, the log barrier is commonly applied [15]. In these cases, the zero solution is naturally precluded. This observation inspires us to propose the following novel formulation, which combines the log barrier with the spectral template (2) to learn graphs without isolated nodes from stationary signals:

$$\min_{\boldsymbol{S}} \ \|\boldsymbol{S}\|_{1,1} - \alpha \boldsymbol{1}^\top \log(\boldsymbol{S1})$$
$$\text{s.t.} \ \ \boldsymbol{S} \in \mathcal{S}, \ \ \boldsymbol{S}\boldsymbol{C}_\infty = \boldsymbol{C}_\infty \boldsymbol{S}, \tag{LogSpecT}$$

where $\|\cdot\|_{1,1}$ is a convex relaxation of $\|\cdot\|_0$ promoting graph sparsity, $\boldsymbol{1}^\top \log(\boldsymbol{S1})$ is the penalty to guarantee the nonexistence of isolated nodes, and $\alpha$ is the tuning parameter. As can be seen from the following proposition, the hyperparameter $\alpha$ in LogSpecT only affects the scale of edge weights instead of the graph connectivity structure.

**Proposition 3.4.** *Let $\text{Opt}(\boldsymbol{C}, \alpha)$ be the optimal solution set of LogSpecT with input covariance matrix $\boldsymbol{C}$ and parameter $\alpha > 0$. Then, for any $\gamma > 0$, it follows that*

$$\text{Opt}(\boldsymbol{C}, \alpha) = \gamma \, \text{Opt}(\boldsymbol{C}, \alpha/\gamma) = \alpha \, \text{Opt}(\boldsymbol{C}, 1).$$

**Remark 3.5.** *The result of Proposition 3.4 spares us from tuning the hyperparameter $\alpha$ when we are coping with binary graphs. In fact, certain normalization will eliminate the impact of different values of $\alpha$ and preserve the connectivity information. Hence, we may simply set $\alpha = 1$ in implementation.*

Note that the true covariance matrix $\boldsymbol{C}_\infty$ in LogSpecT is usually unknown and an estimate $\boldsymbol{C}_n$ from $n$ i.i.d samples is available. To tackle the estimation inaccuracy, we introduce the following robust

formulation:

$$\min_{\boldsymbol{S}} \ \|\boldsymbol{S}\|_{1,1} - \alpha \mathbf{1}^\top \log(\boldsymbol{S}\mathbf{1})$$
$$\text{s.t.} \quad \boldsymbol{S} \in \mathcal{S}, \ \ \|\boldsymbol{S}\boldsymbol{C}_n - \boldsymbol{C}_n\boldsymbol{S}\|_F^2 \leq \delta_n^2. \tag{rLogSpecT}$$

This formulation substitutes $\boldsymbol{C}_\infty$ with $\boldsymbol{C}_n$ and relaxes the equality constraint to an inequality constraint with a tolerance threshold $\delta_n$. Contrary to rSpecT, we prove that rLogSpecT is always feasible.

**Proposition 3.6.** *For $\delta_n > 0$ and $\boldsymbol{C}_n$ with any fixed $n$, rLogSpecT always has a nontrivial feasible solution.*

## 4 Recovery Guarantee of rLogSpecT

In this section, we investigate the non-asymptotic behavior of rLogSpecT when more and more i.i.d samples are collected. For the sake of brevity, we denote $f := \|\cdot\|_{1,1} - \alpha\mathbf{1}^\top \log(\cdot\mathbf{1})$. The theoretical recovery guarantee is as follows:

**Theorem 4.1.** *If $\delta_n \geq 2\rho\|\boldsymbol{C}_n - \boldsymbol{C}_\infty\|$ with given $\rho := \max\{f^*, \alpha m, \alpha m(\log\alpha - 1)\}$, then there exist constants $c_1, \tilde{c}_1, \tilde{c}_2 > 0$ such that*

(i) $|f_n^* - f^*| \leq \varepsilon_n$;   (ii) $\mathrm{dist}(\mathcal{S}^{n,*}, \mathcal{S}_{2\varepsilon_n}^*) \leq \varepsilon_n$;   (iii) $\mathrm{dist}(\mathcal{S}^{n,*}\mathbf{1}, \mathcal{S}_0^*\mathbf{1}) \leq \tilde{c}_1\varepsilon_n + \tilde{c}_2\sqrt{\varepsilon_n}$,

*where $\varepsilon_n := c_1(\delta_n + 2\rho\|\boldsymbol{C}_n - \boldsymbol{C}_\infty\|)$, $\mathcal{S}^{n,*}$ is the optimal solution set of rLogSpecT, $f^*$ (resp. $f_n^*$) denotes the optimal value of LogSpecT (resp. rLogSpecT), and*

$$\mathcal{S}_\varepsilon^* := \{\boldsymbol{S} \in \mathcal{S} \mid \boldsymbol{S}\boldsymbol{C}_\infty = \boldsymbol{C}_\infty\boldsymbol{S}, \ f(\boldsymbol{S}) \leq f^* + \varepsilon\}$$

*is the $\varepsilon$-suboptimal solution set of LogSpecT.*

**Remark 4.2.** (i) *Compared with the conclusion* (ii) *in the Theorem 4.1, the conclusion* (iii) *links the optimal solution sets of rLogSpecT and LogSpecT instead of the sub-optimal solutions.*
(ii) *A byproduct of the proof (see Appendix E.2 for details) shows that the optimal node degree vector set ($\mathcal{S}_0^*\mathbf{1}$) is a singleton. However, there is no guarantee of the uniqueness of $\boldsymbol{S}^*$ itself. We will discuss the impact of such non-uniqueness on model performance in Section 6.4.*

**Remark 4.3.** *The proof of Theorem 4.1 relies on two important optimization concepts: Epi-convergence and truncated Hausdorff distance. Epi-convergence is closely related to the asymptotic solution behavior of approximate minimization problems, and the truncated Hausdorff distance is used to characterize the epi-convergence non-asymptotically. With the help of the Kenmochi condition, which allows us to explicitly calculate the truncated Hausdorff distance, we are able to study the non-asymptotic behavior of the optimal value and optimal solutions of the models. We refer the reader to Appendix E and also [28, Chapter 7], [29, Chapter 6.J] for more details.*

**Corollary 4.4.** *Under the assumptions in Theorem 4.1, it follows that if $\delta_n \to 0$, then*

$$\lim_{n\to\infty} \mathrm{dist}(\mathcal{S}^{n,*}, \mathcal{S}_0^*) = 0. \tag{5}$$

**Remark 4.5.** *From the strong law of large numbers, if we choose $\delta_n = \mathcal{O}(\|\boldsymbol{C}_n - \boldsymbol{C}_\infty\|)$, then almost surely one has $\lim_{n\to\infty} \|\boldsymbol{C}_n - \boldsymbol{C}_\infty\| = 0$ and consequently $\delta_n \to 0$. Hence, Corollary 4.4 shows that (5) holds almost surely.*

For the remaining part of this section, we study the choice of $\delta_n$ under certain statistical assumptions. A large number of distributions (e.g., Gaussian distributions, exponential distributions and any bounded distributions) can be covered by the sub-Gaussian distribution (cf. [41, Section 2.5]), whose formal definition is as follows.

**Definition 4.6** (Sub-Gaussian Distributions). *The probability distribution of a random vector $\boldsymbol{w}$ is called sub-Gaussian if there are $C, v > 0$ such that for every $t > 0$, $\mathbb{P}(\|\boldsymbol{w}\|_2 > t) \leq Ce^{-vt^2}$.*

Consider the case that $\boldsymbol{w}$ in the generative model (1) follows a sub-Gaussian distribution. The following result is adapted from [40, Proposition 2.1].

**Lemma 4.7.** *Suppose that $\boldsymbol{w}$ in the generative model (1) follows a sub-Gaussian distribution. Then, $\boldsymbol{x}$ follows a sub-Gaussian distribution, and with probability larger than $1 - \mathcal{O}(\frac{1}{n})$, one has $\|\boldsymbol{C}_n - \boldsymbol{C}_\infty\| \leq \mathcal{O}(\sqrt{\log n/n})$.*

Equipped with non-asymptotic results in Theorem 4.1, we choose $\delta_n$ with a specific decaying rate.

**Corollary 4.8.** *If the input $\boldsymbol{w}$ in (1) follows a sub-Gaussian distribution and $\delta_n = \mathcal{O}(\sqrt{\log n/n})$, then the assumptions in Theorem 4.1 hold with probability larger than $1 - \mathcal{O}(\frac{1}{n})$.*

Corollary 4.8 together with Theorem 4.1 illustrates the non-asymptotic convergence of optimal function value/suboptimal solution set/optimal node degree vector of rLogSpect from $n$ i.i.d samples to the ideal model LogSpect with the convergence rate $\mathcal{O}(\varepsilon_n)/\mathcal{O}(\varepsilon_n)/\mathcal{O}(\sqrt{\varepsilon_n})$, where $\varepsilon_n = \mathcal{O}(|\boldsymbol{C}_n - \boldsymbol{C}_\infty|) \leq \mathcal{O}(\sqrt{\log n/n})$ for sub-Gaussian $\omega$. This matches the convergence rate $\mathcal{O}(\sqrt{\log n/n})$ of classic spectral template models, e.g., Proposition 2 in [31] and Theorem 2 in [23] for the SpecT model, which shows that LogSpecT is also competitive on recovery rate.

## 5   Linearized ADMM for Solving rLogSpecT

In this section, we design a linearized ADMM algorithm to solve rLogSpecT that admits closed-form solutions for the subproblems. The ADMM-type algorithms have been successfully applied to tackle various graph learning tasks [46, 42].

Firstly, we reformulate rLogSpecT such that it fits the ADMM scheme:

$$
\begin{aligned}
\min_{\boldsymbol{S},\boldsymbol{Z},\boldsymbol{q}} \quad & \langle \boldsymbol{S}, \mathbf{1}\mathbf{1}^\top \rangle - \alpha \mathbf{1}^\top \log \boldsymbol{q} + \iota_{\mathcal{S}}(\boldsymbol{S}) + \iota_{\mathbb{B}(\mathbf{0},\delta_n)}(\boldsymbol{Z}) \\
\text{s.t.} \quad & \boldsymbol{C}_n \boldsymbol{S} - \boldsymbol{S} \boldsymbol{C}_n = \boldsymbol{Z}, \quad \boldsymbol{q} = \boldsymbol{S}\mathbf{1}.
\end{aligned}
\tag{6}
$$

The augmented Lagrangian function of problem (6) is

$$
\begin{aligned}
L(\boldsymbol{S}, \boldsymbol{Z}, \boldsymbol{q}, \boldsymbol{\Lambda}, \boldsymbol{\lambda}_2) = {} & \langle \mathbf{1}\mathbf{1}^\top, \boldsymbol{S} \rangle - \alpha \mathbf{1}^\top \log \boldsymbol{q} + \iota_{\mathcal{S}}(\boldsymbol{S}) + \iota_{\mathbb{B}(\mathbf{0},\delta_n)}(\boldsymbol{Z}) + \langle \boldsymbol{\Lambda}, \boldsymbol{C}_n \boldsymbol{S} - \boldsymbol{S}\boldsymbol{C}_n - \boldsymbol{Z} \rangle \\
& + \boldsymbol{\lambda}_2^\top (\boldsymbol{q} - \boldsymbol{S}\mathbf{1}) + \frac{\rho}{2}\|\boldsymbol{C}_n\boldsymbol{S} - \boldsymbol{S}\boldsymbol{C}_n - \boldsymbol{Z}\|_F^2 + \frac{\rho}{2}\|\boldsymbol{q} - \boldsymbol{S}\mathbf{1}\|_2^2.
\end{aligned}
$$

**Update of primal variables $\boldsymbol{Z}$ and $\boldsymbol{q}$**   Since $\boldsymbol{Z}$ and $\boldsymbol{q}$ are separable given $\boldsymbol{S}$, we update the primal variables in two blocks: $\boldsymbol{S}$ and $(\boldsymbol{Z}, \boldsymbol{q})$. More specifically, the update of $(\boldsymbol{Z}, \boldsymbol{q})$ in the $k$-th iteration is

$$
(\boldsymbol{Z}^{(k+1)}, \boldsymbol{q}^{(k+1)}) := \mathrm{argmin}_{(\boldsymbol{Z},\boldsymbol{q})} L(\boldsymbol{S}^{(k)}, \boldsymbol{Z}, \boldsymbol{q}, \boldsymbol{\Lambda}^{(k)}, \boldsymbol{\lambda}_2^{(k)}),
\tag{7}
$$

which admits the following closed-form solution:

**Proposition 5.1.** *The update (7) can be explicitly rewritten as*

$$
\boldsymbol{Z}^{(k+1)} = \min\left\{1, \frac{\delta_n}{\|\tilde{\boldsymbol{Z}}\|_F}\right\} \tilde{\boldsymbol{Z}}, \quad \boldsymbol{q}^{(k+1)} = \frac{\tilde{\boldsymbol{q}} + \sqrt{\tilde{\boldsymbol{q}}^2 + 4\alpha/\rho\mathbf{1}}}{2},
$$

*where $\tilde{\boldsymbol{Z}} = \boldsymbol{C}_n \boldsymbol{S}^{(k)} - \boldsymbol{S}^{(k)}\boldsymbol{C}_n + \frac{\boldsymbol{\Lambda}^{(k)}}{\rho}$, $\tilde{\boldsymbol{q}} = \boldsymbol{S}^{(k)}\mathbf{1} - \frac{1}{\rho}\boldsymbol{\lambda}_2^{(k)}$.*

**Update of primal variables $\boldsymbol{S}$**   The ADMM update of $\boldsymbol{S}$ does not admit a closed-form solution. Hence, we solve the linearized version $\hat{L}_k$ of the augmented Lagrangian function $L$ when all the variables except $\boldsymbol{S}$ are fixed. It is given by

$$
\hat{L}_k(\boldsymbol{S}) = \langle \boldsymbol{C}^{(k)}, \boldsymbol{S} \rangle + \frac{\rho\tau}{2}\|\boldsymbol{S} - \boldsymbol{S}^{(k)}\|_F^2 + \iota_{\mathcal{S}}(\boldsymbol{S}),
$$

where $\boldsymbol{C}^{(k)} := \mathbf{1}\mathbf{1}^\top + \boldsymbol{C}_n\boldsymbol{\Lambda}^{(k)} - \boldsymbol{\Lambda}^{(k)}\boldsymbol{C}_n - \boldsymbol{\lambda}_2^{(k)}\mathbf{1}^\top + \rho(\boldsymbol{S}^{(k)}\boldsymbol{C}_n^2 + \boldsymbol{C}_n^2\boldsymbol{S}^{(k)} - 2\boldsymbol{C}_n\boldsymbol{S}^{(k)}\boldsymbol{C}_n + \boldsymbol{Z}^{(k+1)}\boldsymbol{C}_n - \boldsymbol{C}_n\boldsymbol{Z}^{(k+1)} - \boldsymbol{q}^{(k+1)}\mathbf{1}^\top + \boldsymbol{S}^{(k)}\mathbf{1}\mathbf{1}^\top)$ and $\tau$ is the parameter for linearization. Therefore, the update of $\boldsymbol{S}$ in the $k$-th iteration is

$$
\boldsymbol{S}^{(k+1)} := \mathrm{argmin}_{\boldsymbol{S} \in \mathcal{S}}\hat{L}_k(\boldsymbol{S}) = \Pi_{\mathcal{S}}(\boldsymbol{S}^{(k)} - \boldsymbol{C}^{(k)}/\rho\tau),
\tag{8}
$$

where $\Pi_{\mathcal{S}}(\boldsymbol{X})$ is the projection of $\boldsymbol{X}$ onto the closed and convex set $\mathcal{S}$ and can be computed by

$$
(\Pi_{\mathcal{S}}(\boldsymbol{X}))_{ij} = \begin{cases} \frac{1}{2}\max\{0, X_{ij} + X_{ji}\}, & \text{if } i \neq j, \\ 0, & \text{if } i = j. \end{cases}
$$

**Update of dual variables and parameters**    Finally, we update the dual variables $\mathbf{\Lambda}$ and $\boldsymbol{\lambda}_2$ via gradient ascent steps of the augmented Lagrangian function:

$$\mathbf{\Lambda}^{(k+1)} := \mathbf{\Lambda}^{(k)} + \rho(\boldsymbol{C}_n \boldsymbol{S}^{(k+1)} - \boldsymbol{S}^{(k+1)} \boldsymbol{C}_n - \boldsymbol{Z}^{(k+1)}), \tag{9}$$

$$\boldsymbol{\lambda}_2^{(k+1)} := \boldsymbol{\lambda}_2^{(k)} + \rho(\boldsymbol{q}^{(k+1)} - \boldsymbol{S}^{(k+1)} \mathbf{1}). \tag{10}$$

Together with the updating rule for the augmented parameter $\rho$, we complete the whole algorithm. The details are presented in Appendix F.

**Complexity of each iteration**    To illustrate the efficiency of L-ADMM, we calculate the complexity of each subproblem. In the update of $\boldsymbol{Z}$ and $\boldsymbol{q}$, the most time consuming part is to calculate $\boldsymbol{C}_n \boldsymbol{S}^{(k)}$. When the number of nodes $m$ is larger than the sample size $n$, we rewrite $\boldsymbol{C}_n \boldsymbol{S}^{(k)}$ as $(1/n\boldsymbol{X})(\boldsymbol{X}^\top \boldsymbol{S}^{(k)})$, where $\boldsymbol{X} \in \mathbb{R}^{m \times n}$ is the data matrix. In this case, the complexity is in the order of $\mathcal{O}(nm^2)$. When $m$ is smaller than $n$, we first calculate $\boldsymbol{C}_n$ and then multiply it with $\boldsymbol{S}^{(k)}$. This gives the complexity $\mathcal{O}(nm^2)$. Similarly, the complexity needed to update $\boldsymbol{S}$ is $\mathcal{O}(nm^2 + mn^2)$. The update of the dual variables can be decomposed as the summation of components calculated when updating the primal variables. In summary, the total complexity of each iteration is $\mathcal{O}(nm^2 + mn^2)$.

**Convergence of L-ADMM**    For the convergence analysis of L-ADMM, we treat $(\boldsymbol{Z}, \boldsymbol{q})$ as one variable, and the two constraints $\boldsymbol{C}_n \boldsymbol{S} - \boldsymbol{S} \boldsymbol{C}_n = \boldsymbol{Z}$ and $\boldsymbol{q} = \boldsymbol{S}\mathbf{1}$ can be written into a single one:

$$[\boldsymbol{I} \otimes \boldsymbol{C}_n - \boldsymbol{C}_n \otimes \boldsymbol{I}; \boldsymbol{I} \otimes \mathbf{1}^\top] \operatorname{vec}(\boldsymbol{S}) = [\operatorname{vec}(\boldsymbol{Z}); \boldsymbol{q}].$$

Then we can apply a two-block proximal ADMM (i.e., L-ADMM) to the problem by alternatingly updating $\boldsymbol{S}$ (see details in Appendix F.4 for this part) and $(\boldsymbol{Z}, \boldsymbol{q})$. Consequently, the convergence result in [45, Theorem 4.2] can be invoked directly to derive the following theorem.

**Theorem 5.2.** *If $\tau > m + \|\boldsymbol{C}_n \otimes \boldsymbol{I}_m - \boldsymbol{I}_m \otimes \boldsymbol{C}_n\|^2$, then $\lim_{k\to\infty} f(\boldsymbol{S}^{(k)}) = f_n^*$.*

## 6    Experiments

In this section, we evaluate our proposed algorithm and models via numerical experiments. The experiments are conducted on both synthetic and real networks. We apply the standard metrics [18]: F-measure, Precision, and Recall to assess the quality of learned graphs. The source code is available at: https://github.com/StevenSYL/NeurIPS2023-LogSpecT.

### 6.1    Data Generation

**Random Graphs**.    In the experiments, we consider two types of synthetic graphs, namely, the Erdős-Rényi (ER) graph [9] and the Barabasi-Albert model graph (BA) [2]. The ER graphs are generated by placing an edge between each pair of nodes independently with probability $p = 0.2$ and the weight on each edge is set to 1. The BA graphs are generated by having two connected nodes initially and then adding new nodes one at a time, where each new node is connected to exactly one previous node that is randomly chosen with a probability proportional to its degree at the time.

**Graph Filters and Signals**.    Three graph filters are used in the experiments. The first one is the low-pass graph filter (lowpass-EXP) $h(\boldsymbol{S}) = \exp(\frac{1}{2}\boldsymbol{S})$. The second one is the high-pass graph filter (highpass-EXP) $h(\boldsymbol{S}) = \exp(-\boldsymbol{S})$. The last one is a quadratic graph filter (QUA) $h(\boldsymbol{S}) = \boldsymbol{S}^2 + \boldsymbol{S} + \boldsymbol{I}$. Note that the quadratic graph filter is neither low-pass nor high-pass. We will also use random graph filters developed from the above ones for experiments on real networks. They will be introduced when applied. The stationary graph signals are generated from the generative model (1). The input graph signal $\boldsymbol{w} \sim \mathcal{N}(\mathbf{0}, \boldsymbol{I}_m)$ is a random vector following the normal distribution.

### 6.2    How to Infer Binary Graphs

Many practical tasks require binary graphs instead of weighted graphs. However, the results from LogSpect and rLogSpecT are generally weighted. In this section, we tackle the issue of how to convert a weighted graph to a binary one. Firstly, we normalize each edge in the graph $\boldsymbol{W}$ by dividing it over the maximal weight, which yields a collection of values ranging from 0 to 1. Our task is then to select a threshold $\varepsilon$ to round these values to 0 or 1 and construct a binary graph $\boldsymbol{W}^*$ accordingly. Mathematically, this procedure is captured by the formula:

$$(W^*)_{ij} = \begin{cases} 1, & \text{if } W_{ij}/\max\{W_{i'j'}\} \geq \varepsilon, \\ 0, & \text{if } W_{ij}/\max\{W_{i'j'}\} < \varepsilon. \end{cases}$$

To choose the threshold $\varepsilon$, we either use a training-based strategy or a searching-based strategy. The training-based one searches the best $\varepsilon$ on $k$ graphs and applies the value to the newly learned graph. The searching-based one simply searches for $\varepsilon$ that yields the best performance for a graph.

### 6.3 Convergence of L-ADMM

We present the convergence performance of L-ADMM in this section. Since there does not literally exist customized algorithm for rLogSpecT, we compare our L-ADMM with CVXPY [8]. The solver used in CVXPY is MOSEK. We conduct the experiments on 100-node BA graphs with 10000 stationary graph signals generated from low-pass EXP graph filter. The parameter $\delta$ is set as $10\sqrt{\log n/n}$. For L-ADMM algorithm, the target accuracy is set as $10^{-6}$ and the initialization is set as zero. The running time for L-ADMM takes around **20** seconds while the solver takes over **110** seconds. The primal residual and dual residual of L-ADMM in each iteration are plot in Figure 1. The results corroborate the efficiency of our proposed algorithm. Furthermore, the locally linear convergence can be observed from the experiment. This implies that L-ADMM converges with fast speed to high accuracy. It remains an open question to prove the property theoretically.

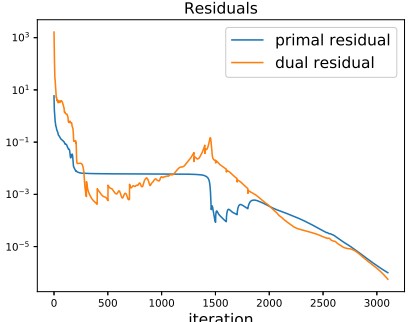

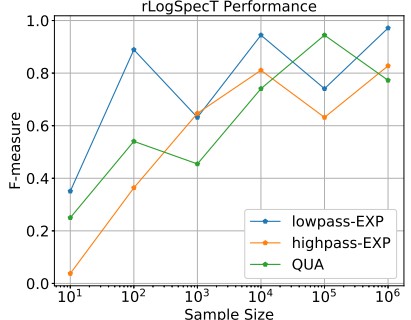

Figure 1: L-ADMM residual on 100-node graph

Figure 2: rLogSpecT with $\delta_n = 0.2\sqrt{\log n/n}$

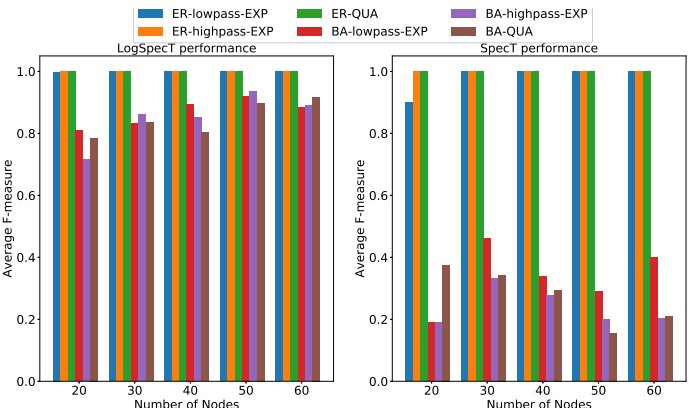

Figure 3: Comparison of LogSpecT & SpecT on Synthetic Data

### 6.4 Experiments on Synthetic Networks

To evaluate the efficacy of LogSpecT and rLogSpecT, we compare them with SpecT and rSpecT on synthetic data. Notice that the prior of no isolated nodes has been incorporated in practice by adding restrictions on the smallest degree. This approach introduces an additional hyperparameter and may exacerbate the infeasibility issue in robust formulation. Also, no recovery guarantees are developed for this trick. For these reasons, we still compare LogSpect & rLogSpecT with SpecT & rSpecT.

**Performance of LogSpecT**. We first compare the performance of the ideal models: LogSpecT and SpecT on the two types of random graphs with three graph filters. We conduct experiments on graphs with nodes ranging from 20 to 60 and use the searching-based strategy to set threshold $\varepsilon^*$. After 10 repetitions, we report the average results in Figure 3. The left column presents the performance of LogSpecT and the right column presents that of SpecT. Firstly, we observe that on ER graphs, both models achieve good performance. They nearly recover the graphs perfectly. However, the performance on BA graphs differs. In this case, LogSpecT can work efficiently. The learned graphs from LogSpecT enjoy much higher F-measure values than those from SpecT. Secondly, the comparison between different graph filters shows that different graph filters have few impacts on recovery performance for both LogSpecT and SpecT. Finally, we observe that the outputs of LogSpecT on BA graphs tend to possess higher F-measure when the number of nodes increases. This suggests that LogSpecT may behave better on larger networks. However, such phenomena cannot be observed from SpecT results.

**Performance of rLogSpecT**. Since LogSpecT persuasively outperforms SpecT, we only present the performance of rLogSpecT when more samples are collected. As we have mentioned before, the optimal solution to LogSpecT is not necessarily unique. Thus, we do not expect to be able to show how rLogSpecT's optimal solution converges to LogSpecT's optimal solution. Moreover, the non-uniqueness may jeopardize the performance of rLogSpecT intuitively. However, the following experiment on BA graphs shows that the learned graphs from rLogSpecT with the binary-graph-transforming strategy tend to approach the ground truth when enough samples are collected.

In this experiment, the sample size $n$ is chosen from 10 to $10^6$ and $\delta_n$ is set as $0.2\sqrt{\log n/n}$. We rely on the training-based strategy to obtain the best threshold $\varepsilon^*$ from 10 randomly chosen training graphs. We then calculate the F-measure of the learned binary graph in the testing set. The result is reported in Figure 2. It shows that rLogSpecT works for all three graph filters on BA graphs and tends to possess higher F-measure values when more and more signals are collected. This indicates that similar to LogSpecT, rLogSpecT is efficient for different types of graph filters on BA graphs, including the high-pass ones. For more experiments and discussions, we refer readers to Appendix G.

### 6.5 Experiments on Real Networks

In this set of experiments, we compare the performance of LogSpecT (resp. rLogSpecT) with SpecT (resp. rSpecT) and other methods from the statistics and GSP communities on *Protein* database and *Reddit* database from [6]. The *Protein* database is a Bioinformatics dataset, where nodes are secondary structure elements (SSEs) and there is an edge between two nodes if they are neighbors in the amino-acid sequence or in 3D space. The *Reddit* database is a social network dataset. In this dataset, each graph corresponds to an online discussion thread, where nodes correspond to users, and there is an edge between two nodes if at least one of them responds to another's comments. We choose graphs in databases whose numbers of nodes are smaller than 50 and generate stationary graph signals from the generative model (1), with the graph filtering chosen as $H(\boldsymbol{S}) = \exp\frac{1}{2}\boldsymbol{S}$. Our experiments include in total 871 testing graphs in the *Protein* database and 309 in *Reddit* database.

**Infeasibility of rSpecT**. For these real networks, we first check whether the infeasibility issue encountered by rSpecT is significant. To this end, we adopt the random graph filters $t_1\boldsymbol{S}^2 + t_2\boldsymbol{S} + t_3\boldsymbol{I}$, where $t_i, i = 1, 2, 3$ are random variables following a Gaussian distribution with $\mu = 0$ and $\sigma = 2$. Then, we calculate the smallest $\delta_n$ such that rSpecT is feasible. If the smallest $\delta_n > 0$, rSpecT is likely to be encountered with the infeasibility issue. The results with different numbers of graph signals observed are shown in Table 1.

Table 1: Likelihood of infeasibility

|  | Protein | | | Reddit | | |
|---|---|---|---|---|---|---|
| sample size | 10 | 100 | 1000 | 10 | 100 | 1000 |
| frequency | 0.980 | 0.999 | 0.994 | 0.984 | 1 | 1 |
| mean of $\delta_{\min}$ | 38.514 | 26.012 | 14.813 | 1094.788 | 830.642 | 531.006 |

The experiment results present a decrease in the mean value of $\delta_{\min}$ when more samples are used. However, a high frequency of the infeasibility issue occurring in these real datasets cannot be mitigated by the increasing sample size. These observations highlight the necessity of the careful treatment for choosing $\delta$ in rSpecT again.

**Performance of different graph learning models**. In this section, the stationary graph signals are generated by the low-pass-EXP filter. We choose the famous statistical method called (thresholded) correlation [17] and the first GSP method that applies the log barrier to graph inference [15] as the baselines. The optimal threshold for the correlation method is selected from $0.1$ to $0.6$ and we search in $\{0.01, 0.1, 1, 10, 100, 1000\}$ to obtain the best hyperparameter in Kalofolias' model. The parameter $\delta_n$ in rLogSpecT is set as $10\sqrt{\log n/n}$ and in rSpecT it is set as the smallest value that allows a feasible solution [31]. We also rely on the searching-based strategy to convert the learned weighted graphs from (r)SpecT & (r)LogSpecT.

The results of the ideal models with the true covariance matrix $C_\infty$ applied are collected in Figure 4. We observe that on the real graphs, LogSpecT achieves the best performance on average (median represented by the red line). Also, compared with SpecT, LogSpecT performs more stably. We remark that since the graph signals are not necessarily smooth, Kalofolias' model cannot provide guaranteed performance, especially on the Reddit networks.

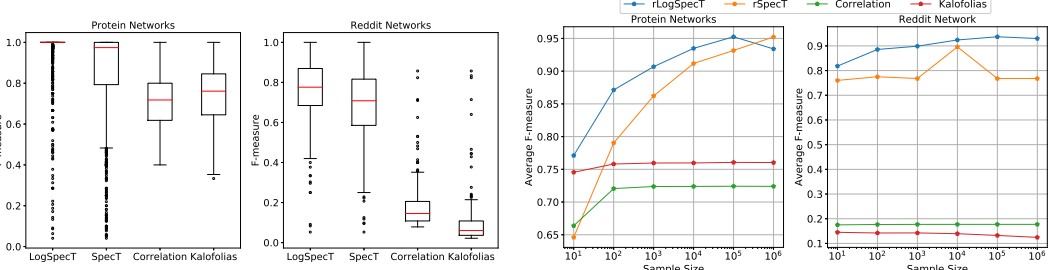

Figure 4: Performance of LogSpecT and SpecT   Figure 5: Asymptotic behavior of rLogSpecT and rSpecT.

Figure 5 compares the performance of four different methods when different numbers of signals are observed.[2] When the sample size increases, the models focusing on stationarity property can recover the graphs more accurately while correlation method and Kalofolias' method fail to exploit the signal information. This can also be inferred from the experiment results in Figure 4 since the models fail to achieve good performance from full information, let alone from partial information. The experiment also shows that when a fixed number of samples are observed, the learned graphs from rLogSpecT approximate the ground truth better than rSpecT. This further corroborates the superiority of rLogSpecT on graph learning from stationary signals.

# 7   Conclusion

In this paper, we directly tackle the infeasibility issue encountered in graph learning from stationary graph signals [31] and propose the first condition that guarantees the model infeasibility. To overcome this, we propose an efficient alternative. The recovery guarantees of its robust formulation are analyzed with advanced optimization tools, which may find broader applications in learning tasks. Compared with current literature [23], these theoretical results require less stringent conditions. We also design an L-ADMM algorithm that allows for efficient implementation and theoretical convergence. Extensive experiments on both synthetic and real data are conducted in this paper. The results show that our proposed model can significantly outperform the existing ones.

We believe this work represents an important step beyond the fundamental model SpecT. Its general formulation allows for the transfer of the current extensions made on SpecT. Testing our proposed models with these extensions is one future direction. Also, we notice that although the recovery guarantees for robust formulations are clear, the estimation performance analysis for the ideal-case models (i.e. SpecT and LogSpecT) is still incomplete. Investigating the exact recovery conditions is another future direction.

---

[2]The model in [34] is a substitute of rSpecT to approximate SpecT and its hyperparameter is hard to determine. Hence, we omit the performance of that model.

## Acknowledgments and Disclosure of Funding

This work was supported in part by the Hong Kong Research Grants Council (RGC) General Research Fund (GRF) Project CUHK 14203920.

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

## A  Organization of the Appendix

The appendix includes the missing proofs, detailed discussions of some argument in the main body and more numerical experiments. We organize the appendix as follows:

- The proof of infeasibility condition (Theorem 3.2) is provided in Section B.
- Explanations on conditions derived in Theorem 3.2 are included in Section C.
- The proof of properties of the proposed model (r)LogSpecT (Proposition 3.4 & 3.6) is given in Section D and some additional properties are discussed.
- The truncated Hausdorff distance based proof details of Theorem 4.1 and Corollary 4.4 are given in Section E.
- Details of L-ADMM and its convergence analysis are in Section F.
- Additional experiments and discussions on synthetic and real data are included in Section G.

## B  Proof of Theorem 3.2

Since the linear system (4) has no solution, we know from Farkas' lemma that the following system has solutions:

$$\begin{cases} \begin{bmatrix} \boldsymbol{I}_{m-1} & \boldsymbol{0}_{\frac{(m-1)(m-2)}{2}} \end{bmatrix} \boldsymbol{B}^\top \boldsymbol{A}_n^\top \boldsymbol{x} < \boldsymbol{0}_{(m-1)\times 1}, \\ \begin{bmatrix} \boldsymbol{0}_{\frac{(m-1)(m-2)}{2}\times(m-1)} & \boldsymbol{I}_{\frac{(m-1)(m-2)}{2}} \end{bmatrix} \boldsymbol{B}^\top \boldsymbol{A}_n^\top \boldsymbol{x} \leq \boldsymbol{0}_{\frac{(m-1)(m-2)}{2}\times 1}. \end{cases} \tag{11}$$

Let $\boldsymbol{x}^* \in \mathbb{R}^{m^2}$ be a solution to (11). Denote $\boldsymbol{x}_+ := \max\{\boldsymbol{x}^*, \boldsymbol{0}\}$, $\boldsymbol{x}_- := \max\{-\boldsymbol{x}^*, \boldsymbol{0}\}$. Then, there exists $c \in (0,1]$ such that

$$\boldsymbol{B}^\top \boldsymbol{A}_n^\top (\boldsymbol{x}_+ - \boldsymbol{x}_-) + c \boldsymbol{1}_{m^2}^\top (\boldsymbol{x}_+ + \boldsymbol{x}_-)[\boldsymbol{1}_{m-1}; \boldsymbol{0}_{\frac{(m-1)(m-2)}{2}}] \leq \boldsymbol{0}.$$

Define $y := -\boldsymbol{1}_{m^2}^\top (\boldsymbol{x}_+ + \boldsymbol{x}_-)$, $z := c\boldsymbol{1}_{m^2}^\top (\boldsymbol{x}_+ + \boldsymbol{x}_-)$ and set $\bar{\delta} = c$. For all $\delta \in [0, \bar{\delta})$, $(\boldsymbol{x}_+, \boldsymbol{x}_-, y, z)$ is a solution to the following linear system:

$$\begin{cases} \boldsymbol{B}^\top \boldsymbol{A}_n^\top (\boldsymbol{x}_+ - \boldsymbol{x}_-) + z[\boldsymbol{1}_{m-1}; \boldsymbol{0}_{\frac{(m-1)(m-2)}{2}}] \leq \boldsymbol{0}, \\ \boldsymbol{1}_{m^2}^\top (\boldsymbol{x}_+ + \boldsymbol{x}_-) + y \leq 0, \\ \delta y + z > 0, \\ \boldsymbol{x}_+, \boldsymbol{x}_-, -y \geq \boldsymbol{0}. \end{cases}$$

Again, from Farkas' lemma, this implies that the following linear system does not have a solution:

$$\begin{cases} \boldsymbol{A}_n \boldsymbol{B} \boldsymbol{s} + t\boldsymbol{1}_{m^2} \geq \boldsymbol{0}, \\ \boldsymbol{A}_n \boldsymbol{B} \boldsymbol{s} - t\boldsymbol{1}_{m^2} \leq \boldsymbol{0}, \\ t \leq \delta, \\ \begin{bmatrix} \boldsymbol{1}_{m-1} & \boldsymbol{0}_{\frac{(m-1)(m-2)}{2}} \end{bmatrix} \boldsymbol{s} = 1, \end{cases} \tag{12}$$

where $\boldsymbol{s} \in \mathbb{R}^{m(m-1)/2}$ and $t \in \mathbb{R}$. Since (12) is equivalent to:

$$\begin{cases} \|\boldsymbol{C}_n \boldsymbol{S} - \boldsymbol{S} \boldsymbol{C}_n\|_{\infty,\infty} \leq \delta, \\ (\boldsymbol{S}\boldsymbol{1})_1 = 1, \\ \boldsymbol{S} \in \mathcal{S}, \end{cases} \tag{13}$$

the above argument indicates that (13) does not have a solution. Suppose rSpecT has a feasible solution $\boldsymbol{S}'$, then

$$\|\boldsymbol{C}_n \boldsymbol{S}' - \boldsymbol{S}' \boldsymbol{C}_n\|_{\infty,\infty} \leq \|\boldsymbol{C}_n \boldsymbol{S}' - \boldsymbol{S}' \boldsymbol{C}_n\|_F \leq \delta.$$

Hence, $\boldsymbol{S}'$ is also a solution to (13). However, (13) does not have a solution. We can conclude that rSpecT is infeasible in this case.

## C   Explanations on Sufficient Conditions in Theorem 3.2

We elaborate more on the infeasibility condition that $A_n B$ has full column rank. An application of the condition is Example 3.1. Specifically, we know that in this case,

$$
B = \begin{pmatrix} 0 \\ 1 \\ 1 \\ 0 \end{pmatrix} \quad \text{and} \quad A_n = \begin{pmatrix} 0 & h_{12} & -h_{12} & 0 \\ h_{12} & h_{22} - h_{11} & 0 & -h_{12} \\ -h_{12} & 0 & h_{11} - h_{22} & h_{12} \\ 0 & -h_{12} & h_{12} & 0 \end{pmatrix}.
$$

This implies that

$$
A_n B = \begin{pmatrix} 0 \\ h_{22} - h_{11} \\ h_{11} - h_{22} \\ 0 \end{pmatrix}.
$$

Hence, when $h_{11} \neq h_{22}$, $A_n B$ has full column rank. This means that when $\delta$ is small enough (from Example 3.1 we know $\tilde{\delta} = \sqrt{2}|h_{11} - h_{22}|$), the model rSpecT is infeasible.

## D   Proofs of Properties of (r)LogSpecT

### D.1   Proof of Proposition 3.4

Since the constraint set $\mathcal{S}$ is a cone, it follows that for all $\gamma > 0$, $\gamma \mathcal{S} = \mathcal{S}$. Then, we know that

$$
\begin{aligned}
\mathrm{Opt}(C, \alpha) &= \underset{S \in \mathcal{S}, CS = SC}{\mathrm{argmin}} \|S\|_{1,1} - \alpha \mathbf{1}^\top \log(S\mathbf{1}) \\
&= \gamma \cdot \underset{\gamma S \in \mathcal{S}, C\gamma S = \gamma SC}{\mathrm{argmin}} \|\gamma S\|_{1,1} - \alpha \mathbf{1}^\top \log(\gamma S\mathbf{1}) \\
&= \gamma \cdot \underset{S \in \frac{1}{\gamma}\mathcal{S}, CS = SC}{\mathrm{argmin}} \gamma\|S\|_{1,1} - \alpha \mathbf{1}^\top \log(S\mathbf{1}) \\
&= \gamma \cdot \underset{S \in \mathcal{S}, CS = SC}{\mathrm{argmin}} \|S\|_{1,1} - \frac{\alpha}{\gamma} \mathbf{1}^\top \log(S\mathbf{1}) \\
&= \gamma \, \mathrm{Opt}(C, \alpha/\gamma),
\end{aligned}
$$

where the third equality is from the basic calculus rule of the logarithm function. Set $\gamma = \alpha$ and then $\mathrm{Opt}(C, \alpha) = \alpha \, \mathrm{Opt}(C, 1)$, which completes the proof.

### D.2   Proof of Proposition 3.6

The proof will be conducted by constructing a feasible solution for rLogSpecT. Recall that $A_n = I \otimes C_n - C_n \otimes I$ and the matrix $B \in \mathbb{R}^{m^2 \times m(m-1)/2}$ that maps a non-negative vector to the vectorization of a valid adjacency matrix. Let $S = \min\{\frac{\delta}{\|A_n Bs\|_2}, 1\} \cdot \mathrm{mat}(Bs)$ with $s \in \mathbb{R}^{(m-1)m/2}$ being a non-negative vector, where $\mathrm{mat}(\cdot)$ is the matricization operator. Note that

$$
\mathrm{vec}(C_n S - S C_n) = (I \otimes C_n - C_n \otimes I)\,\mathrm{vec}(S) = A_n \mathrm{vec}(S).
$$

Then, we know that

$$
\|C_n S - S C_n\|_F = \|\mathrm{vec}(C_n S - S C_n)\|_2 = \min\left\{\frac{\delta}{\|A_n Bs\|_2}, 1\right\} \cdot \|A_n Bs\|_2 \leq \delta.
$$

Thus, the given $S$ is a feasible solution for rLogSpecT and it completes the proof.

### D.3   Properties of optimal solutions and values of (r)LogSpecT

In this section, we further discuss some properties of the optimal solutions/value of the proposed models, which are useful for deriving the recovery guarantee. More specifically, we obtain an upper bound on the optimal solutions (which may not be unique) independent of the sample size $n$ and the inaccuracy parameter $\delta_n$. Also, a lower bound of optimal values follows.

**Proposition D.1.** *The following statements hold:*

- *For an optimal solution $\boldsymbol{S}^*$ (resp. $\boldsymbol{S}_n^*$) to LogSpecT (resp. rLogSpecT with any given sample size $n$), it follows that*

$$\|\boldsymbol{S}^*\|_{1,1} = \alpha m \ \text{ and } \ \|\boldsymbol{S}_n^*\|_{1,1} \leq \alpha m, \ \ \forall \delta_n > 0.$$

- *If $\delta_n \geq 2\alpha m \|\boldsymbol{C}_n - \boldsymbol{C}_\infty\|$, then*

$$\alpha m (1 - \log \alpha) \leq f_n^* \leq f^*, \ \ \forall n,$$

*where $f^*$ (resp. $f_n^*$) denotes the optimal value of LogSpecT (resp. rLogSpecT).*

For the first statement, let us consider the Karush-Kuhn-Tucker (KKT) conditions of LogSpecT and rLogSpecT. Since the LogSpecT is a convex problem and Slater's condition holds, the KKT conditions are necessary and sufficient for the optimality, i.e., there exists $(\boldsymbol{\Lambda}_1, \boldsymbol{\Lambda}_2) \in \mathbb{R}^{m \times m} \times \mathcal{N}_\mathcal{S}(\boldsymbol{S}^*)$ such that

$$\begin{cases} \nabla_{\boldsymbol{S}}(\|\boldsymbol{S}^*\|_{1,1} - \alpha \mathbf{1}^\top \log(\boldsymbol{S}^*\mathbf{1})) + \boldsymbol{C}_\infty \boldsymbol{\Lambda}_1 - \boldsymbol{\Lambda}_1 \boldsymbol{C}_\infty + \boldsymbol{\Lambda}_2 = \mathbf{0}, \\ \boldsymbol{C}_\infty \boldsymbol{S}^* = \boldsymbol{S}^* \boldsymbol{C}_\infty, \\ \boldsymbol{S}^* \in \mathcal{S}, \end{cases} \tag{14}$$

where $\mathcal{N}_\mathcal{S}(\boldsymbol{S}^*) := \{\boldsymbol{N} \in \mathbb{R}^{m \times m} : \sup_{\boldsymbol{X} \in \mathcal{S}} \langle \boldsymbol{X} - \boldsymbol{S}^*, \boldsymbol{N} \rangle \leq 0\}$ is the normal cone of $\mathcal{S}$ at $\boldsymbol{S}^*$, and $\nabla \|\boldsymbol{S}^*\|_{1,1}$ is well-defined since $\|\cdot\|_{1,1} = \langle \cdot, \mathbf{11}^\top \rangle$ at $\boldsymbol{S}^* \geq 0$, which is differentiable. Taking further calculation gives that

$$\nabla \|\boldsymbol{S}^*\|_{1,1} = \mathbf{11}^\top, \quad (\nabla_{\boldsymbol{S}} \mathbf{1}^\top \log(\boldsymbol{S}^*\mathbf{1}))_{ij} = \frac{1}{(\boldsymbol{S}^*\mathbf{1})_i}.$$

Combining this with (14) by taking inner product of both sides with $\boldsymbol{S}^*$, we obtain that

$$\sum_{i,j}(\boldsymbol{S}^*)_{ij} - \alpha \sum_{i,j} \frac{(\boldsymbol{S}^*)_{ij}}{(\boldsymbol{S}^*\mathbf{1})_i} + \langle \boldsymbol{\Lambda}_1, \boldsymbol{C}_\infty \boldsymbol{S}^* - \boldsymbol{S}^* \boldsymbol{C}_\infty \rangle + \langle \boldsymbol{\Lambda}_2, \boldsymbol{S}^* \rangle = 0. \tag{15}$$

From the structure of $\mathcal{S}$ and the fact that $\boldsymbol{\Lambda}_2 \in \mathcal{N}_\mathcal{S}(\boldsymbol{S}^*)$, one has that $\langle \boldsymbol{\Lambda}_2, \boldsymbol{S}^* \rangle = 0$. Also, note that $\boldsymbol{C}_\infty \boldsymbol{S}^* = \boldsymbol{S}^* \boldsymbol{C}_\infty$. Hence, the equation (15) can be simplified as the desired result:

$$\|\boldsymbol{S}^*\|_{1,1} = \sum_{i,j}(\boldsymbol{S}^*)_{ij} = \alpha \sum_{i,j} \frac{(\boldsymbol{S}^*)_{ij}}{(\boldsymbol{S}^*\mathbf{1})_i} = \alpha \sum_{i=1}^m \sum_{j=1}^m \frac{(\boldsymbol{S}^*)_{ij}}{(\boldsymbol{S}^*\mathbf{1})_i} = \alpha m.$$

The KKT conditions of rLogSpecT indicate that there exist $\lambda_1 \geq 0$, $\boldsymbol{\Lambda}_2 \in \mathcal{N}_\mathcal{S}(\boldsymbol{S}_n^*)$ and $\boldsymbol{Q} \in \partial \|\boldsymbol{C}_n \boldsymbol{S}_n^* - \boldsymbol{S}_n^* \boldsymbol{C}_n\|_F$ (i.e., the subgradient of the function $\boldsymbol{S} \mapsto \|\boldsymbol{C}_n \boldsymbol{S} - \boldsymbol{S} \boldsymbol{C}_n\|_F$ at $\boldsymbol{S}_n^*$) such that

$$\begin{cases} \nabla_{\boldsymbol{S}}(\|\boldsymbol{S}_n^*\|_{1,1} - \alpha \mathbf{1}^\top \log(\boldsymbol{S}_n^*\mathbf{1})) + \lambda_1 \boldsymbol{Q} + \boldsymbol{\Lambda}_2 = \mathbf{0}, \\ \lambda_1(\|\boldsymbol{C}_n \boldsymbol{S}_n^* - \boldsymbol{S}_n^* \boldsymbol{C}_n\|_F - \delta_n) = 0, \\ \boldsymbol{S}_n^* \in \mathcal{S}. \end{cases} \tag{16}$$

Moreover, from the definition of the convex subdifferential we know that $0 \geq \|\boldsymbol{C}_n \boldsymbol{S}_n^* - \boldsymbol{S}_n^* \boldsymbol{C}_n\|_F - \langle \boldsymbol{Q}, \boldsymbol{S}_n^* \rangle$. Thus, after taking inner product of both sides of the equation (16) with $\boldsymbol{S}_n^*$, it follows that:

$$\begin{aligned} 0 &= \sum_{i,j}(\boldsymbol{S}_n^*)_{ij} - \alpha m + \lambda_1 \langle \boldsymbol{Q}, \boldsymbol{S}_n^* \rangle + \langle \boldsymbol{\Lambda}_2, \boldsymbol{S}_n^* \rangle \\ &\geq \sum_{i,j}(\boldsymbol{S}_n^*)_{ij} - \alpha m + \lambda_1 \|\boldsymbol{C}_n \boldsymbol{S}_n^* - \boldsymbol{S}_n^* \boldsymbol{C}_n\|_F + \langle \boldsymbol{\Lambda}_2, \boldsymbol{S}_n^* \rangle \\ &= \sum_{i,j}(\boldsymbol{S}_n^*)_{ij} - \alpha m + \lambda_1 \delta_n, \end{aligned}$$

which implies that $\sum_{i,j}(\boldsymbol{S}_n^*)_{ij} \leq \alpha m - \lambda_1 \delta_n \leq \alpha m$. This completes the proof of the first statement.

For the second statement, we first prove that $v_n^*$ and $v^*$ are larger than $\alpha m(1 - \log \alpha)$. Define the auxiliary function $g : \mathbb{R} \to \mathbb{R}$ such that $g(x) := x - \alpha \log x$ for any $x \in \mathbb{R}_+$, whose minimum is attained at $\alpha$. Since for any $\boldsymbol{S} \in \mathcal{S}$,

$$f(\boldsymbol{S}) = \sum_{i=1}^{m} g\left(\sum_{j=1}^{m} S_{ij}\right),$$

where $f$ is the objective in LogSpecT, it follows that

$$f(\boldsymbol{S}) \geq \sum_{i=1}^{m} g(\alpha) = \alpha m(1 - \log \alpha).$$

This implies that $v_n^*$ and $v^*$ are larger than $\alpha m(1 - \log \alpha)$. Next, we will show $v_n^* \leq v^*$. Consider any optimal solution $\boldsymbol{S}^*$ to LogSpecT. We show that it is feasible for rLogSpecT.

$$\begin{aligned}
\|\boldsymbol{C}_n \boldsymbol{S}^* - \boldsymbol{S}^* \boldsymbol{C}_n\| &= \|\boldsymbol{C}_n \boldsymbol{S}^* - \boldsymbol{C}_\infty \boldsymbol{S}^* + \boldsymbol{S}^* \boldsymbol{C}_\infty - \boldsymbol{S}^* \boldsymbol{C}_n\| \\
&\leq 2\|\boldsymbol{S}^*\|_{1,1}\|\boldsymbol{C}_n - \boldsymbol{C}_\infty\| \leq 2\alpha m\|\boldsymbol{C}_n - \boldsymbol{C}_\infty\| \leq \delta_n,
\end{aligned}$$

where the equality comes from $\boldsymbol{C}_\infty \boldsymbol{S}^* = \boldsymbol{S}^* \boldsymbol{C}_\infty$, the first inequality comes from the fact that $\|\boldsymbol{X}\boldsymbol{Y}\| \leq \|\boldsymbol{X}\|_F \|\boldsymbol{Y}\| \leq \|\boldsymbol{X}\|_{1,1}\|\boldsymbol{Y}\|$, the second one comes from the first statement and the last one is due to $\delta_n \geq 2\alpha m\|\boldsymbol{C}_n - \boldsymbol{C}_\infty\|$. Hence, $\boldsymbol{S}^*$ is feasible for rLogSpecT, which indicates that $v_n^* \leq v^*$. The proof is completed.

# E   Proof of Theorem 4.1 & Corollary 4.4

## E.1   Truncated Hausdorff distance

In this section, we introduce an advanced technique in optimization that is efficient in analyzing the recovery guarantee of robust formulations. Before that, we introduce the concept of truncated Hausdorff distance between two sets.

**Definition E.1** (Truncated Hausdorff Distance [29, 6.J]). *For any $\rho \geq 0$, the truncated Hausdorff distance between two sets $\mathcal{C}$ and $\mathcal{D}$ is defined as:*

$$\hat{\mathrm{d}}_\rho(\mathcal{C}, \mathcal{D}) = \max\{\mathrm{dist}(\mathcal{C} \cap \mathbb{B}(\boldsymbol{0}, \rho), \mathcal{D}), \mathrm{dist}(\mathcal{D} \cap \mathbb{B}(\boldsymbol{0}, \rho), \mathcal{C})\}.$$

It turns out that the distance between the optimum of two minimization problems can be bounded with the truncated Hausdorff distance of the epigraphs under some conditions. The result is captured in the following lemma.

**Lemma E.2** ([29, Theorem 6.56]). *Let $\rho \in [0, \infty)$. Suppose that the extended-real-valued lower semicontinuous functions $f, g : \mathbb{R}^n \to \overline{\mathbb{R}}$ satisfy*

- *$\inf f, \inf g \in [-\rho, \rho]$,*

- *$\mathrm{argmin}\, f, \mathrm{argmin}\, g \subseteq \mathbb{B}(\boldsymbol{0}, \rho)$.*

*Then, it follows that*

$$|\inf f - \inf g| \leq \hat{\mathrm{d}}_\rho(\mathrm{epi}\, f, \mathrm{epi}\, g).[3] \tag{17}$$

*Suppose further that $\varepsilon \geq 2\,\hat{\mathrm{d}}_\rho(\mathrm{epi}\, f, \mathrm{epi}\, g)$, then one has*

$$\mathrm{dist}(\boldsymbol{x}_g^*, \varepsilon\text{-}\mathrm{argmin}\, f) \leq \hat{\mathrm{d}}_\rho(\mathrm{epi}\, f, \mathrm{epi}\, g), \tag{18}$$

*where $\varepsilon$-$\mathrm{argmin}\, f$ is the $\varepsilon$-suboptimal solution set of $f$ that is defined as $\varepsilon$-$\mathrm{argmin}\, f := \{\boldsymbol{x} \in \mathrm{dom}\, f : f(\boldsymbol{x}) \leq \inf f + \varepsilon\}$, and $\boldsymbol{x}_g^*$ is a minimizer of $g$.*

From the above lemma, we know that if two optimization problems are close enough (in the sense of truncated Hausdorff distance), then the optimum of them should be close to each other. Hence, in order to apply this result, we need to bound the truncated Hausdorff distance in an explicit way, which is solved by the following Kenmochi condition.

---

[3] For a function $f : \mathbb{R}^n \to \overline{\mathbb{R}}$, its epigraph is defined as $\mathrm{epi}\, f := \{(\boldsymbol{x}, y) \mid y \geq f(\boldsymbol{x})\}$.

**Lemma E.3** (Kenmochi Condition [29, Proposition 6.58]). *Let $\rho \in [0, \infty)$. Then, for $f, g : \mathbb{R}^n \to \overline{\mathbb{R}}$ with nonempty epigraphs, one has that*

$$\hat{\mathrm{d}}_\rho(\mathrm{epi}\, f, \mathrm{epi}\, g) = \inf \left\{ \eta > 0 : \begin{array}{l} \inf\limits_{\mathbb{B}(\boldsymbol{x}, \eta)} g \leq \max\{f(\boldsymbol{x}), -\rho\} + \eta, \; \forall \boldsymbol{x} \in [f \leq \rho] \cap \mathbb{B}(\boldsymbol{0}, \rho) \\ \inf\limits_{\mathbb{B}(\boldsymbol{x}, \eta)} f \leq \max\{g(\boldsymbol{x}), -\rho\} + \eta, \; \forall \boldsymbol{x} \in [g \leq \rho] \cap \mathbb{B}(\boldsymbol{0}, \rho) \end{array} \right\},$$

*where $[f \leq \rho] := \{\boldsymbol{x} \in \mathbb{R}^n : f(\boldsymbol{x}) \leq \rho\}$.*

### E.2 Proof of Theorem 4.1

Before presenting the proof, we first introduce the following lemma.

**Lemma E.4** (Hoffman's Error Bound [12]). *Consider the set $\mathcal{S} := \{\boldsymbol{x} \in \mathbb{R}^n : \boldsymbol{Ax} \leq \boldsymbol{b}\}$. There exists $C > 0$ such that for any $\boldsymbol{x} \in \mathbb{R}^n$, one has*

$$\mathrm{dist}(\boldsymbol{x}, \mathcal{S}) \leq C \cdot \|(\boldsymbol{Ax} - \boldsymbol{b})_+\|_2.$$

For the sake of brevity, we denote

$$\bar{f}_n(\boldsymbol{S}) := \|\boldsymbol{S}\|_{1,1} - \alpha \mathbf{1}^\top \log(\boldsymbol{S1}) + \iota_{\mathbb{R}_-}(\|\boldsymbol{C}_n\boldsymbol{S} - \boldsymbol{S}\boldsymbol{C}_n\|_F - \delta_n) + \iota_{\mathcal{S}}(\boldsymbol{S}),$$
$$\bar{f}(\boldsymbol{S}) := \|\boldsymbol{S}\|_{1,1} - \alpha \mathbf{1}^\top \log(\boldsymbol{S1}) + \iota_{\{0\}}(\|\boldsymbol{C}_\infty\boldsymbol{S} - \boldsymbol{S}\boldsymbol{C}_\infty\|_F) + \iota_{\mathcal{S}}(\boldsymbol{S}).$$

Hence, the optimization problem LogSpecT (resp. rLogSpecT) is equivalent to $\inf \bar{f}$ (resp. $\inf \bar{f}_n$).

Now, we aim to use Lemma E.3 to bound $\hat{\mathrm{d}}_\rho(\mathrm{epi}\, \bar{f}, \mathrm{epi}\, \bar{f}_n)$. Let $\boldsymbol{S} \in \mathcal{S} \cap \mathbb{B}(\boldsymbol{0}, \rho)$ satisfy

$$\bar{f}(\boldsymbol{S}) \leq \rho \quad \text{and} \quad \boldsymbol{S}\boldsymbol{C}_\infty = \boldsymbol{C}_\infty\boldsymbol{S}.$$

Then, we know that

$$\|\boldsymbol{S}\boldsymbol{C}_n - \boldsymbol{C}_n\boldsymbol{S}\|_F \leq 2\|\boldsymbol{S}\|_F \|\boldsymbol{C}_n - \boldsymbol{C}_\infty\| \leq 2\rho\|\boldsymbol{C}_n - \boldsymbol{C}_\infty\| \leq \delta_n,$$

and consequently $\boldsymbol{S}$ is in the domain of $\bar{f}_n$. Then, it follows that for any $\eta > 0$, we have

$$\inf_{\mathbb{B}(\boldsymbol{S}, \eta)} \bar{f}_n \leq \bar{f}_n(\boldsymbol{S}) = \bar{f}(\boldsymbol{S}) \leq \max\{\bar{f}(\boldsymbol{S}), -\rho\}, \quad \forall \boldsymbol{S} \in [\bar{f} \leq \rho] \cap \mathbb{B}(\boldsymbol{0}, \rho). \tag{19}$$

Before verifying the reverse side of the Kenmochi condition, we first consider the non-emptiness of $[\bar{f}_n \leq \rho] \cap \mathbb{B}(\boldsymbol{0}, \rho)$. Since

$$\delta_n \geq 2\rho\|\boldsymbol{C}_n - \boldsymbol{C}_\infty\| \geq 2\alpha m\|\boldsymbol{C}_n - \boldsymbol{C}_\infty\|,$$

it follows from Proposition D.1 that $\|\boldsymbol{S}_n^*\|_{1,1} \leq \alpha m \leq \rho$ and $f_n^* \leq f^* \leq \rho$, which implies that $[\bar{f}_n \leq \rho] \cap \mathbb{B}(\boldsymbol{0}, \rho)$ is nonempty. Let $\boldsymbol{S}_n \in [\bar{f}_n \leq \rho] \cap \mathbb{B}(\boldsymbol{0}, \rho)$. Then, one has that

$$\boldsymbol{S}_n \in \mathcal{S} \quad \text{and} \quad \|\boldsymbol{C}_n\boldsymbol{S}_n - \boldsymbol{S}_n\boldsymbol{C}_n\|_F \leq \delta_n.$$

Hence, it follows that

$$\|\boldsymbol{C}_\infty\boldsymbol{S}_n - \boldsymbol{S}_n\boldsymbol{C}_\infty\| \leq 2\|\boldsymbol{S}_n\|_F \|\boldsymbol{C}_\infty - \boldsymbol{C}_n\| + \|\boldsymbol{C}_n\boldsymbol{S}_n - \boldsymbol{S}_n\boldsymbol{C}_n\|_F \leq 2\rho\|\boldsymbol{C}_\infty - \boldsymbol{C}_n\| + \delta_n.$$

Also, note that there exists $\beta > 0$ such that $(\boldsymbol{S}_n\mathbf{1})_i \geq \beta$ for all $i \in [m]$ as $\bar{f}_n \leq \rho$ and $\|\boldsymbol{S}_n\|_{1,1} - \alpha \mathbf{1}^\top \log(\boldsymbol{S}_n\mathbf{1}) \to \infty$ when $\boldsymbol{S}_n \to \boldsymbol{0}$. Thus, applying Lemma E.4 to the linear system

$$\tilde{\mathcal{S}} := \{\boldsymbol{S} \in \mathbb{R}^{m \times m} : \boldsymbol{S}\boldsymbol{C}_\infty = \boldsymbol{C}_\infty\boldsymbol{S}, \; \boldsymbol{S} \in \mathcal{S}, \; (\boldsymbol{S1})_i \geq \beta, \; \forall i \in [m]\}$$

yields that there exists $\tilde{c} > 0$ such that

$$\mathrm{dist}(\boldsymbol{S}_n, \tilde{\mathcal{S}}) \leq \tilde{c} \cdot (2\rho\|\boldsymbol{C}_\infty - \boldsymbol{C}_n\| + \delta_n).$$

Hence, there exists $\tilde{\boldsymbol{S}}$ in the domain of $\bar{f}$ such that

$$\|\boldsymbol{S}_n - \tilde{\boldsymbol{S}}\|_F \leq \tilde{c} \cdot (2\rho\|\boldsymbol{C}_\infty - \boldsymbol{C}_n\| + \delta_n) \quad \text{and} \quad (\tilde{\boldsymbol{S}}\mathbf{1})_i \geq \beta, \; \forall i \in [m].$$

Since the function $\boldsymbol{S} \mapsto \|\boldsymbol{S}\|_{1,1} - \alpha \mathbf{1}^\top \log(\boldsymbol{S}\mathbf{1})$ is locally Lipschitz continuous when $(\boldsymbol{S}\mathbf{1})_i \geq \beta$, there exists $L > 0$ such that

$$
\begin{aligned}
\bar{f}(\tilde{\boldsymbol{S}}) = \|\tilde{\boldsymbol{S}}\|_{1,1} - \alpha \mathbf{1}^\top \log(\tilde{\boldsymbol{S}}\mathbf{1}) &\leq \|\boldsymbol{S}_n\|_{1,1} - \alpha \mathbf{1}^\top \log(\boldsymbol{S}_n\mathbf{1}) + L\|\boldsymbol{S}_n - \tilde{\boldsymbol{S}}\|_F \\
&= \bar{f}_n(\boldsymbol{S}_n) + L\|\boldsymbol{S}_n - \tilde{\boldsymbol{S}}\|_F \\
&\leq \bar{f}_n(\boldsymbol{S}_n) + L\tilde{c} \cdot (2\rho\|\boldsymbol{C}_\infty - \boldsymbol{C}_n\| + \delta_n).
\end{aligned}
$$

Setting $c_1 \geq \max\{1, L\} \cdot \tilde{c}$, one can obtain that for any $\boldsymbol{S}_n \in [\bar{f}_n \leq \rho] \cap \mathbb{B}(\mathbf{0}, \rho)$

$$
\inf_{\mathbb{B}(\boldsymbol{S}_n, \eta)} \bar{f} \leq \bar{f}(\tilde{\boldsymbol{S}}) \leq \bar{f}_n(\boldsymbol{S}_n) + c_1 \cdot (2\rho\|\boldsymbol{C}_\infty - \boldsymbol{C}_n\| + \delta_n) \leq \max\{\bar{f}_n(\boldsymbol{S}_n), -\rho\} + \eta, \quad (20)
$$

where $\eta := c_1 \cdot (2\rho\|\boldsymbol{C}_\infty - \boldsymbol{C}_n\| + \delta_n)$. Combining inequality (19) and (20), we can conclude that

$$
\hat{\mathrm{d}}_\rho(\mathrm{epi}\,\bar{f}, \mathrm{epi}\,\bar{f}_n) \leq c_1 \cdot (2\rho\|\boldsymbol{C}_\infty - \boldsymbol{C}_n\| + \delta_n). \quad (21)
$$

In order to derive the conclusion (i) and (ii), it remains to check the requirements in Lemma E.2. Since $\rho \geq \alpha m$, the first statement of Proposition D.1 shows that the optimal solutions to $\inf \bar{f}$ and $\inf \bar{f}_n$ lie in $\mathbb{B}(\mathbf{0}, \rho)$. Since $\rho \geq f^*$ and $-\rho \leq \alpha m(1 - \log \alpha)$, the second statement of the proposition shows that $\inf \bar{f}, \inf \bar{f}_n \in [-\rho, \rho]$. Hence, applying Lemma E.2 completes the proof of the first two statements.

To prove conclusion (iii), we first make the following two claims:

    (a) $\mathcal{S}_0^*\mathbf{1}$ is a singleton, whose element is denoted by $\boldsymbol{S}^*\mathbf{1}$,

    (b) For any $\bar{\varepsilon} \in [0, \infty)$, there exists a $\delta(\bar{\varepsilon}) > 0$ such that for all $0 \leq \varepsilon \leq \bar{\varepsilon}$ and $\boldsymbol{S}_\varepsilon \in \mathcal{S}_\varepsilon^*$, one has that

$$
\|\boldsymbol{S}_\varepsilon\mathbf{1} - \boldsymbol{S}^*\mathbf{1}\|_2 \leq \delta(\bar{\varepsilon}) \cdot \sqrt{\varepsilon}. \quad (22)
$$

Granting these and with the help of Theorem 4.1 (ii), we can derive that for all $\boldsymbol{S}_n^* \in \mathcal{S}^{n,*}$

$$
\begin{aligned}
\mathrm{dist}(\boldsymbol{S}_n^*\mathbf{1}, \mathcal{S}_0^*\mathbf{1}) = \|\boldsymbol{S}_n^*\mathbf{1} - \boldsymbol{S}^*\mathbf{1}\|_2 &\leq \|\boldsymbol{S}_n^*\mathbf{1} - \boldsymbol{S}_{2\varepsilon_n}\mathbf{1}\|_2 + \|\boldsymbol{S}_{2\varepsilon_n}\mathbf{1} - \boldsymbol{S}^*\mathbf{1}\|_2 \\
&\leq \sqrt{m}\,\mathrm{dist}(\boldsymbol{S}_n^*, \mathcal{S}_{2\varepsilon_n}^*) + \|\boldsymbol{S}_{2\varepsilon_n}\mathbf{1} - \boldsymbol{S}^*\mathbf{1}\|_2 \\
&\leq \tilde{c}_1\varepsilon_n + \tilde{c}_2\sqrt{\varepsilon_n},
\end{aligned}
$$

where $\tilde{c}_1, \tilde{c}_2$ are positive constants, and $\boldsymbol{S}_{2\varepsilon_n} \in \mathcal{S}_{2\varepsilon_n}^*$ satisfies $\|\boldsymbol{S}_n^* - \boldsymbol{S}_{2\varepsilon_n}\|_F = \mathrm{dist}(\boldsymbol{S}_n^*, \mathcal{S}_{2\varepsilon_n}^*)$ (whose existence is guaranteed since $\mathcal{S}_\varepsilon^*$ is convex and compact). Hence,

$$
\mathrm{dist}(\mathcal{S}^{n,*}\mathbf{1}, \mathcal{S}_0^*\mathbf{1}) \leq \tilde{c}_1\varepsilon_n + \tilde{c}_2\sqrt{\varepsilon_n}.
$$

To proceed, it remains to prove the claims. Define an auxiliary function $h : \mathbb{R}^m \to \mathbb{R}$ as $h(\boldsymbol{x}) = \sum_{i=1}^m x_i - \alpha \sum_{i=1}^m \log x_i$ for each $\boldsymbol{x} \in \mathbb{R}_+^m$. Consider the following optimization problem:

$$
\begin{aligned}
\min_{\boldsymbol{x}} \quad & h(\boldsymbol{x}) \\
\text{s.t.} \quad & \boldsymbol{x} \in \{\boldsymbol{S}\mathbf{1} \in \mathbb{R}^m \mid \boldsymbol{S} \text{ that is feasible for LogSpecT}\}.
\end{aligned} \quad (23)
$$

For the sake of brevity, denote the $\varepsilon$-suboptimal solution set of (23) as $\mathcal{H}_\varepsilon^*$. In the remaining part, we will first show that $\mathcal{S}_\varepsilon^*\mathbf{1} = \mathcal{H}_\varepsilon^*$ and then, by the strict convexity of $h$, the desired two claims hold.

The first step is to show that the optimal function value of the problem (23) satisfies $h^* = f^*$. Since it is obvious that $\tilde{\boldsymbol{x}} = \boldsymbol{S}^*\mathbf{1}$ is feasible for (23), $h^* \leq h(\tilde{\boldsymbol{x}}) = f(\boldsymbol{S}^*) = f^*$. Suppose to the contrary that $h^* < f^*$, from the fact that the objective function is coercive and continuous and the feasible set is closed, there exists $\tilde{\boldsymbol{S}}$ such that it is feasible for LogSpecT and $\boldsymbol{x}^* = \tilde{\boldsymbol{S}}\mathbf{1}$, where $\boldsymbol{x}^*$ is an optimal solution to (23). Since $h^* = h(\boldsymbol{x}^*) = h(\tilde{\boldsymbol{S}}\mathbf{1}) = f(\tilde{\boldsymbol{S}})$, this contradicts the fact that $f(\tilde{\boldsymbol{S}}) \geq f^*$. Hence, $h^* = f^*$. Next, we will show that $\mathcal{S}_\varepsilon^*\mathbf{1} = \mathcal{H}_\varepsilon^*$. Consider any $\varepsilon$-suboptimal solution $\boldsymbol{S} \in \mathcal{S}_\varepsilon^*$, i.e.,

$$
h(\boldsymbol{S}\mathbf{1}) = f(\boldsymbol{S}) \leq f^* + \varepsilon = h^* + \varepsilon.
$$

Hence, $\boldsymbol{S}\mathbf{1} \in \mathcal{H}_\varepsilon^*$ and it implies that $\mathcal{S}_\varepsilon^*\mathbf{1} \subseteq \mathcal{H}_\varepsilon^*$. On the other hand, for any $\varepsilon$-suboptimal solution $\boldsymbol{x} \in \mathcal{H}_\varepsilon^*$, there exists $\boldsymbol{S}$ that is feasible for LogSpecT such that $\boldsymbol{x} = \boldsymbol{S}\mathbf{1}$. Thus,

$$
f(\boldsymbol{S}) = h(\boldsymbol{x}) \leq h^* + \varepsilon = f^* + \varepsilon.
$$

This implies that $\boldsymbol{S} \in \mathcal{S}_{\varepsilon}^*$ and consequently $\mathcal{H}_{\varepsilon}^* \subseteq \mathcal{S}_{\varepsilon}^* \mathbf{1}$. Hence, $\mathcal{H}_{\varepsilon}^* = \mathcal{S}_{\varepsilon}^* \mathbf{1}$.

Since $h$ is strictly convex, its optimal solution set $\mathcal{H}_0^*$ is a singleton. Then, $\mathcal{S}_0^* \mathbf{1} = \mathcal{H}_0^*$ is a singleton, which proves the first claim. For the second claim, we know that for any $\boldsymbol{S}_\varepsilon \in \mathcal{S}_\varepsilon^*$ there exists $\boldsymbol{x}_\varepsilon \in \mathcal{H}_\varepsilon^*$ such that

$$\|\boldsymbol{S}_\varepsilon \mathbf{1} - \boldsymbol{S}^* \mathbf{1}\|_2 = \|\boldsymbol{x}_\varepsilon - \boldsymbol{x}^*\|_2, \tag{24}$$

where $\boldsymbol{x}^* \in \mathcal{H}_0^*$. The coerciveness of $h$ asserts that $\boldsymbol{x}_\varepsilon$ and $\boldsymbol{x}^*$ are bounded. This together with the fact that $h$ is strongly convex on any bounded set, illustrates that there exists $\mu > 0$ such that

$$h(\boldsymbol{x}_\varepsilon) \ge h(\boldsymbol{x}^*) + \langle \nabla h(\boldsymbol{x}^*), \boldsymbol{x}_\varepsilon - \boldsymbol{x}^* \rangle + \frac{1}{\mu} \|\boldsymbol{x}_\varepsilon - \boldsymbol{x}^*\|_2^2 \ge h(\boldsymbol{x}^*) + \frac{1}{\mu} \|\boldsymbol{x}_\varepsilon - \boldsymbol{x}^*\|_2^2, \tag{25}$$

where the second inequality comes from the global optimality of $\boldsymbol{x}^*$. Combining (24) and (25) gives that

$$\|\boldsymbol{S}_\varepsilon \mathbf{1} - \boldsymbol{S}^* \mathbf{1}\|_2 = \|\boldsymbol{x}_\varepsilon - \boldsymbol{x}^*\|_2 \le \sqrt{\mu(h(\boldsymbol{x}_\varepsilon) - h(\boldsymbol{x}^*))} \le \sqrt{\mu \varepsilon}.$$

This completes the proof of the claims.

### E.3 Proof of Corollary 4.4

Suppose to the contrary that there exists a sequence $\{\boldsymbol{S}_n^*\}_n$, where the $n$th element is an optimal solution to rLogSpecT with sample size $n$, such that

$$\mathrm{dist}(\boldsymbol{S}_n^*, \mathcal{S}_0^*) \nrightarrow 0.$$

From Proposition D.1, we know that $\{\boldsymbol{S}_n^*\}_n$ is bounded, and consequently, has a convergent subsequence. Without loss of generality, we may assume that the sequence itself is convergent and the limiting point is $\boldsymbol{S}^*$. Note that

$$\|\boldsymbol{C}_n \boldsymbol{S}_n^* - \boldsymbol{S}_n^* \boldsymbol{C}_n\|_F \le \delta_n, \quad \boldsymbol{C}_n \to \boldsymbol{C}_\infty \text{ and } \delta_n \to 0.$$

Hence, $\boldsymbol{C}_\infty \boldsymbol{S}^* = \boldsymbol{S}^* \boldsymbol{C}_\infty$. This indicates that $\boldsymbol{S}^*$ is feasible for LogSpecT. Then, from Theorem 4.1, we know that $f(\boldsymbol{S}_n^*) = f_n^* \to f^*$, which leads to $f(\boldsymbol{S}^*) = f^*$ since $f(\cdot) = \|\cdot\|_{1,1} - \alpha \mathbf{1}^\top \log(\cdot \mathbf{1})$ is continuous. Together with the fact that $\boldsymbol{S}^*$ is feasible, we conclude that $\boldsymbol{S}^*$ is an optimal solution to LogSpecT. This further implies that $\mathrm{dist}(\boldsymbol{S}_n^*, \mathcal{S}_0^*) \to 0$, which is a contradiction.

### E.4 Proof of Lemma 4.7

Recall the generative model (1). Since $\boldsymbol{w}$ follows a sub-Gaussian distribution, it can be shown that for every $t > 0$,

$$\mathbb{P}(\|\boldsymbol{x}\|_2 > t) \le \mathbb{P}\left(\|\boldsymbol{w}\|_2 > \frac{t}{\|\mathcal{H}(\boldsymbol{S})\|}\right) \le C e^{-v' t^2},$$

for some positive constant $v'$, which means that $\boldsymbol{x}$ also follows a sub-Gaussian distribution. Thus, due to the sub-Gaussian property, $\|\boldsymbol{C}_n - \boldsymbol{C}_\infty\|$ can be explictly bounded by the following lemma.

**Lemma E.5** ([40, Proposition 2.1]). *Consider sub-Gaussian, identical, independent random vectors* $\boldsymbol{x}_1, \boldsymbol{x}_2, \dots, \boldsymbol{x}_n \in \mathbb{R}^m$ *with* $n > m$. *Then for all* $\varepsilon > 0$, *it follows that*

$$\mathbb{P}\left(\left\|\frac{1}{n}\sum_{i=1}^n \boldsymbol{x}_i \boldsymbol{x}_i^\top - \mathbb{E}[\boldsymbol{x}\boldsymbol{x}^\top]\right\|_2 \le \varepsilon\right) \ge 1 - 2e^{2m - l\varepsilon^2 n},$$

*for some constant* $l > 0$.

Setting $\varepsilon^2 = (4/l) \log(2n) m/n$, Lemma E.5 indicates that with high probability (lower bounded by $1 - n^{-1}$),

$$\|\boldsymbol{C}_n - \boldsymbol{C}_\infty\| \le \mathcal{O}\left(\sqrt{\frac{\log n}{n}}\right).$$

## F  Derivations of L-ADMM and Convergence Analysis

This section includes the details of L-ADMM for rLogSpecT.

### F.1 Proof of Proposition 5.1

Note that the minimization problem (7) is separable for $\boldsymbol{Z}$ and $\boldsymbol{q}$, and can be split into two subproblems:

$$\min_{\boldsymbol{Z} \in \mathbb{B}(\mathbf{0}, \delta_n)} \|\boldsymbol{C}_n \boldsymbol{S}^{(k)} - \boldsymbol{S}^{(k)} \boldsymbol{C}_n + \boldsymbol{\Lambda}^{(k)}/\rho - \boldsymbol{Z}\|_F^2, \tag{26}$$

$$\min_{\boldsymbol{q}} -\alpha \mathbf{1}^\top \log \boldsymbol{q} + \boldsymbol{\lambda}_2^{(k)\top}(\boldsymbol{q} - \boldsymbol{S}^{(k)}\mathbf{1}) + \frac{\rho}{2}\|\boldsymbol{q} - \boldsymbol{S}^{(k)}\mathbf{1}\|_2^2. \tag{27}$$

For problem (26), the optimal solution is the projection of $\boldsymbol{C}_n \boldsymbol{S}^{(k)} - \boldsymbol{S}^{(k)} \boldsymbol{C}_n + \boldsymbol{\Lambda}^{(k)}/\rho$ onto $\mathbb{B}(\mathbf{0}, \delta_n)$, which is given by

$$\boldsymbol{Z}^{(k+1)} = \min\left\{1, \frac{\delta_n}{\|\tilde{\boldsymbol{Z}}\|_F}\right\} \tilde{\boldsymbol{Z}} \ \text{ with } \ \tilde{\boldsymbol{Z}} = \boldsymbol{C}_n \boldsymbol{S}^{(k)} - \boldsymbol{S}^{(k)} \boldsymbol{C}_n + \boldsymbol{\Lambda}^{(k)}/\rho.$$

For problem (27), the first-order optimality condition gives

$$-\alpha 1/\boldsymbol{q} + \boldsymbol{\lambda}_2^{(k)} + \rho(\boldsymbol{q} - \boldsymbol{S}^{(k)}\mathbf{1}) = 0.$$

This together with the fact that the objective function is convex implies that

$$\boldsymbol{q}^{(k+1)} = \frac{\tilde{\boldsymbol{q}} + \sqrt{\tilde{\boldsymbol{q}}^2 + 4\alpha/\rho \mathbf{1}}}{2} \ \text{ with } \ \tilde{\boldsymbol{q}} = \frac{1}{\rho}(\rho \boldsymbol{S}^{(k)}\mathbf{1} - \boldsymbol{\lambda}_2^{(k)}).$$

### F.2 Calculation of $\Pi_{\mathcal{S}}(\cdot)$

The projection of $\boldsymbol{X}$ to $\mathcal{S}$ can be calculated via an optimization problem:

$$\min_{\boldsymbol{S}} \ \|\boldsymbol{X} - \boldsymbol{S}\|_F^2$$
$$\text{s.t.} \ \ \boldsymbol{S}^\top = \boldsymbol{S},$$
$$S_{ii} = 0, \ i = 1, 2, \ldots, m,$$
$$S_{ij} \geq 0, \ \forall i, j,$$

which is equivalent to

$$\min \ \sum_{i<j}\left((X_{ij} - S_{ij})^2 + (X_{ji} - S_{ij})^2\right)$$
$$\text{s.t.} \ \ S_{ij} \geq 0, \ \forall i < j,$$
$$S_{ii} = 0, \ \forall i.$$

Hence

$$(\Pi_{\mathcal{S}}(\boldsymbol{X}))_{ij} = \begin{cases} \frac{1}{2}\max\{0, X_{ij} + X_{ji}\}, & i \neq j, \\ 0, & i = j. \end{cases}$$

### F.3 Stopping criterion and updating rule of $\rho$

We follow the procedures in [3] to update $\rho$ in each iteration. Similarly, we define the primal residual and dual residual as follows:

$$p_{\text{res}}^{(k+1)} = \sqrt{\|\boldsymbol{Z}^{(k+1)} - \boldsymbol{C}_n\boldsymbol{S}^{(k+1)} + \boldsymbol{S}^{(k+1)}\boldsymbol{C}_n\|_F^2 + \|\boldsymbol{q}^{(k+1)} - \boldsymbol{S}^{(k+1)}\mathbf{1}\|_2^2},$$

$$d_{\text{res}}^{(k+1)} = \rho^{(k)}\left(\boldsymbol{C}_n(\boldsymbol{S}^{(k+1)} - \boldsymbol{S}^{(k)}) - (\boldsymbol{S}^{(k+1)} - \boldsymbol{S}^{(k)})\boldsymbol{C}_n + \mathbf{1}^\top(\boldsymbol{S}^{(k+1)} - \boldsymbol{S}^{(k)})\mathbf{1}\right).$$

The aim of updating $\rho$ is to control the decaying speed of $p_{\text{res}}$ and $d_{\text{res}}$ such that their difference is not too large. To this end, we update $\rho$ adaptively following the scheme:

$$\rho^{(k+1)} := \begin{cases} 2\rho^{(k)}, & \text{if } p_{\text{res}}^{(k+1)} > 5d_{\text{res}}^{(k+1)}, \\ \rho^{(k)}/2, & \text{if } d_{\text{res}}^{(k+1)} > 5p_{\text{res}}^{(k+1)}, \\ \rho^{(k)}, & \text{otherwise.} \end{cases}$$

When $p_{\text{res}}$ and $d_{\text{res}}$ are both smaller than the threshold $\varepsilon = 10^{-5}$, we stop the algorithm.

### F.4 Convergence analysis

Define $\boldsymbol{D} := \mathrm{Diag}(\mathbf{1}_m^\top, \dots, \mathbf{1}_m^\top) \in \mathbb{R}^{m \times m^2}$. Then, $\boldsymbol{D}$ satisfies $\boldsymbol{D}\mathrm{vec}(\boldsymbol{S}) = \boldsymbol{S}\mathbf{1}$ and $\|\boldsymbol{D}^\top \boldsymbol{D}\| = m$. Denote

$$\boldsymbol{Q} := \tau \boldsymbol{I} - \boldsymbol{D}^\top \boldsymbol{D} - \boldsymbol{A}_n^\top \boldsymbol{A}_n.$$

Then the linearized ADMM update (8) of $\boldsymbol{S}$ can be written as:

$$\min_{\boldsymbol{S}} \ L(\boldsymbol{S}) + \frac{\rho}{2} \|\mathrm{vec}(\boldsymbol{S}) - \mathrm{vec}(\boldsymbol{S}^{(k)})\|_{\boldsymbol{Q}},$$

where $\|\boldsymbol{x}\|_{\boldsymbol{Q}} := \boldsymbol{x}^\top \boldsymbol{Q}\boldsymbol{x}$. Since $\tau > m + \|\boldsymbol{A}_n\|^2$, we know that $\boldsymbol{Q}$ is positively definite. Consequently, by treating $(\boldsymbol{Z}, \boldsymbol{q})$ as one variable, we can apply Theorem 4.2 in [45] and directly obtain the result.

## G More Experiments and Discussions on Synthetic and Real Data

### G.1 Influence of graph filters

To make a fair comparison between rSpecT and rLogSpecT, we test rSpecT on BA graphs with the same graph filters and the results are reported in Figure 6. It is obvious that rSpecT fails in these cases and cannot benefit from the increase in sample size. This is reasonable since SpecT fails on BA graphs as indicated in Figure 3, let alone the approximation formulation rSpecT.

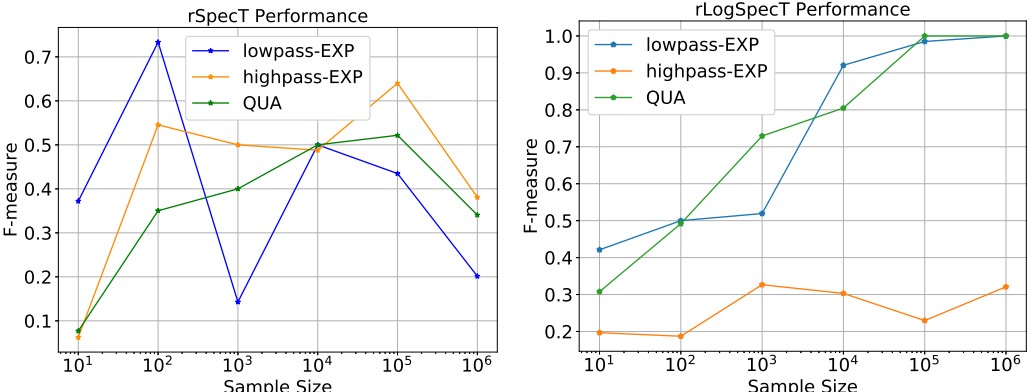

Figure 6: Performance of rSpecT on BA graphs. Figure 7: rLogSpecT on ER graphs with $\delta_n = 20\sqrt{\log n / n}$.

We further test rLogSpecT on ER graphs with different numbers of signals observed. The parameter $\delta_n$ is set as $20\sqrt{\log n / n}$ and the results are reported in Figure 7. The figure shows that for graph filters that are not high-pass, rLogSpecT can achieve nearly perfect recovery when the sample size is large enough. Also, compared with the performance on BA graphs, rLogSpecT works better on ER graphs. This observation is in accordance with the conclusion from Figure 3 that LogSpecT performs better on ER graphs than BA ones. We further notice that the difference between the low-pass graph filter and the high-pass one is huge. To check the conjecture that rLogSpecT generally performs better on low-pass graph filters, we choose different graph filters $\exp(t\boldsymbol{S})$ with $t$ ranging from $-2$ to $2$ and conduct the experiments on ER graphs. When the graph shifting operator is the adjacency matrix, the positive low-pass parameter $t$ corresponds to low-pass graph filters and the negative $t$ corresponds to the high-pass ones [26, 11]. We omit the case when $t = 0$ since this filter does not contain any graph information (note that $\exp(0\boldsymbol{S}) = \boldsymbol{I}$).

We then repeat the experiments for 50 times and report the average results in Figure 8. The comparison between the performance of low-pass graph filters and high-pass graph filters indicates that the low-pass graph filters generally outperforms the high-pass ones. A closer look at the results shows that the performance grows faster when the absolute value of $t$ is smaller. And eventually, the graph filter with smaller absolute value of $t$ prevails. This observation is interesting since Figure 3 indicates that the choice of graph filters has few impacts on the model performance. One explanation is that both low-pass graph filters and high-pass graph filters attenuate some frequencies of the graph and the larger absolute value of $t$ leads to the more loss of information carried by finite signals.

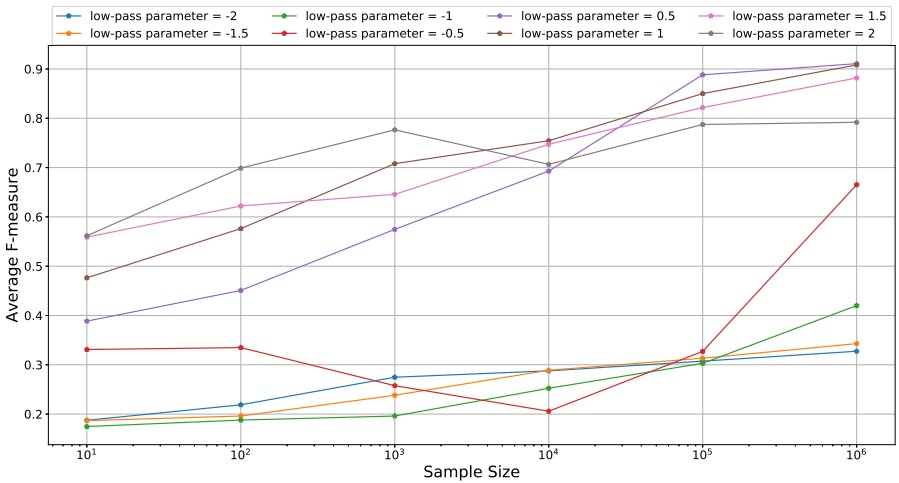

Figure 8: Effect of Low-Pass Parameter: different performance of graph filters $\exp(t\boldsymbol{S})$ with $t$ ranging from $-2$ to $2$.

### G.2 Experiments on USPS dataset

USPS dataset is a handwritten digit dataset. As shown in [25], it is nearly stationary with respect to the nearest neighbour graph. This dataset collects images of handwritten digits. In the experiment, each pixel is viewed as a node and the value on the pixel is view as the graph signal. We follow the approach in [25] to construct the 20 nearest neighbour graph, on which the data is verified to be nearly stationary. More specifically, we pick the 1296 digit 1 images. The weights between two nodes are decided by the Gaussian radial basis function. It turns out that the stationarity measure $s := \|\operatorname{diag}(\boldsymbol{U}^\top \boldsymbol{C}_n \boldsymbol{U})\|_2 / \|\boldsymbol{U}^\top \boldsymbol{C}_n \boldsymbol{U}\|_F$ equals 0.78 in this dataset. Here $\boldsymbol{U}$ is the eigen-matrix of the constructed graph and $\boldsymbol{C}_n$ is the covariance matrix of the observed graph signals. We view the 20 nearest neighbour graph as the groundtruth and compare the learned graph from rLogSpecT with it. We present the subgraphs consisting of the top 10 nodes with the largest degrees in the two graphs respectively in Figure 9. The F-measure of these two subgraphs is 0.96. This result corroborates the efficiency of our proposed rLogSpecT model.

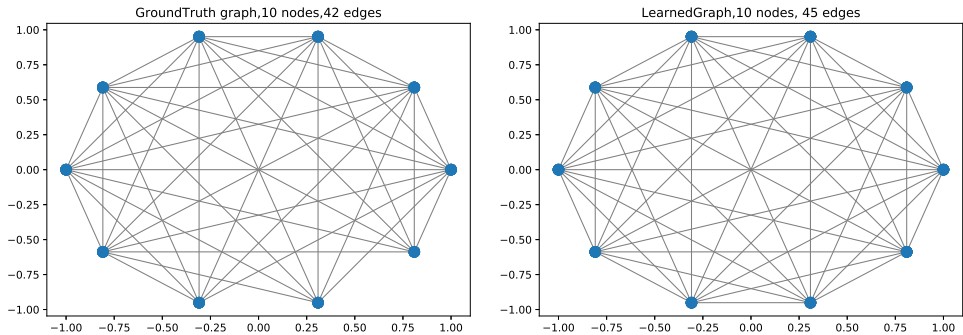

Figure 9: Graph Learning on USPS dataset. (**Left**: subgraph of the symmetric 20 nearest neighbour graph. **Right**: subgraph of the learned graph from rLogSpecT. The F-measure is 0.96)

