## A  Organization of the Appendix

The appendix includes the missing proofs, detailed discussions of some argument in the main body and more numerical experiments. We organize the appendix as follows:

- The proof of infeasibility condition (Theorem 3.2) is provided in Section B.
- Explanations on conditions derived in Theorem 3.2 are included in Section C.
- The proof of properties of the proposed model (r)LogSpecT (Proposition 3.4 & 3.6) is given in Section D and some additional properties are discussed.
- The truncated Hausdorff distance based proof details of Theorem 4.1 and Corollary 4.4 are given in Section E.
- Details of L-ADMM and its convergence analysis are in Section F.
- Additional experiments and discussions on synthetic data are included in Section G.

## B  Proof of Theorem 3.2

Since the linear system (4) has no solution, we know from Farkas' lemma that the following system has solutions:

$$
\begin{cases}
\begin{bmatrix} \boldsymbol{I}_{m-1} & \boldsymbol{0}_{\frac{(m-1)(m-2)}{2}} \end{bmatrix} \boldsymbol{B}^\top \boldsymbol{A}_n^\top \boldsymbol{x} < \boldsymbol{0}_{(m-1)\times 1}, \\
\begin{bmatrix} \boldsymbol{0}_{\frac{(m-1)(m-2)}{2} \times (m-1)} & \boldsymbol{I}_{\frac{(m-1)(m-2)}{2}} \end{bmatrix} \boldsymbol{B}^\top \boldsymbol{A}_n^\top \boldsymbol{x} \leq \boldsymbol{0}_{\frac{(m-1)(m-2)}{2} \times 1}.
\end{cases}
\tag{11}
$$

Let $\boldsymbol{x}^* \in \mathbb{R}^{m^2}$ be a solution to (11). Denote $\boldsymbol{x}_+ := \max\{\boldsymbol{x}^*, \boldsymbol{0}\}$, $\boldsymbol{x}_- := \max\{-\boldsymbol{x}^*, \boldsymbol{0}\}$. Then, there exists $c \in (0,1]$ such that

$$
\boldsymbol{B}^\top \boldsymbol{A}_n^\top (\boldsymbol{x}_+ - \boldsymbol{x}_-) + c\boldsymbol{1}_{m^2}^\top (\boldsymbol{x}_+ + \boldsymbol{x}_-)[\boldsymbol{1}_{m-1}; \boldsymbol{0}_{\frac{(m-1)(m-2)}{2}}] \leq \boldsymbol{0}.
$$

Define $y := -\boldsymbol{1}_{m^2}^\top (\boldsymbol{x}_+ + \boldsymbol{x}_-)$, $z := c\boldsymbol{1}_{m^2}^\top (\boldsymbol{x}_+ + \boldsymbol{x}_-)$ and set $\bar{\delta} = c$. For all $\delta \in [0, \bar{\delta})$, $(\boldsymbol{x}_+, \boldsymbol{x}_-, y, z)$ is a solution to the following linear system:

$$
\begin{cases}
\boldsymbol{B}^\top \boldsymbol{A}_n^\top (\boldsymbol{x}_+ - \boldsymbol{x}_-) + z[\boldsymbol{1}_{m-1}; \boldsymbol{0}_{\frac{(m-1)(m-2)}{2}}] \leq \boldsymbol{0}, \\
\boldsymbol{1}_{m^2}^\top (\boldsymbol{x}_+ + \boldsymbol{x}_-) + y \leq 0, \\
\delta y + z > 0, \\
\boldsymbol{x}_+, \boldsymbol{x}_-, -y \geq \boldsymbol{0}.
\end{cases}
$$

Again, from Farkas' lemma, this implies that the following linear system does not have a solution:

$$
\begin{cases}
\boldsymbol{A}_n \boldsymbol{B}\boldsymbol{s} + t\boldsymbol{1}_{m^2} \geq \boldsymbol{0}, \\
\boldsymbol{A}_n \boldsymbol{B}\boldsymbol{s} - t\boldsymbol{1}_{m^2} \leq \boldsymbol{0}, \\
t \leq \delta, \\
\begin{bmatrix} \boldsymbol{1}_{m-1} & \boldsymbol{0}_{\frac{(m-1)(m-2)}{2}} \end{bmatrix} \boldsymbol{s} = 1,
\end{cases}
\tag{12}
$$

where $\boldsymbol{s} \in \mathbb{R}^{m(m-1)/2}$ and $t \in \mathbb{R}$. Since (12) is equivalent to:

$$
\begin{cases}
\|\boldsymbol{C}_n \boldsymbol{S} - \boldsymbol{S}\boldsymbol{C}_n\|_{\infty,\infty} \leq \delta, \\
(\boldsymbol{S}\boldsymbol{1})_1 = 1, \\
\boldsymbol{S} \in \mathcal{S},
\end{cases}
\tag{13}
$$

the above argument indicates that (13) does not have a solution. Suppose rSpecT has a feasible solution $\boldsymbol{S}'$, then

$$
\|\boldsymbol{C}_n \boldsymbol{S}' - \boldsymbol{S}'\boldsymbol{C}_n\|_{\infty,\infty} \leq \|\boldsymbol{C}_n \boldsymbol{S}' - \boldsymbol{S}'\boldsymbol{C}_n\|_F \leq \delta.
$$

Hence, $\boldsymbol{S}'$ is also a solution to (13). However, (13) does not have a solution. We can conclude that rSpecT is infeasible in this case.

## C   Explanations on Sufficient Conditions in Theorem 3.2

We elaborate more on the infeasibility condition that $\boldsymbol{A}_n\boldsymbol{B}$ has full column rank. An application of the condition is Example 3.1. Specifically, we know that in this case,

$$\boldsymbol{B} = \begin{pmatrix} 0 \\ 1 \\ 1 \\ 0 \end{pmatrix} \quad \text{and} \quad \boldsymbol{A}_n = \begin{pmatrix} 0 & h_{12} & -h_{12} & 0 \\ h_{12} & h_{22} - h_{11} & 0 & -h_{12} \\ -h_{12} & 0 & h_{11} - h_{22} & h_{12} \\ 0 & -h_{12} & h_{12} & 0 \end{pmatrix}.$$

This implies that

$$\boldsymbol{A}_n\boldsymbol{B} = \begin{pmatrix} 0 \\ h_{22} - h_{11} \\ h_{11} - h_{22} \\ 0 \end{pmatrix}.$$

Hence, when $h_{11} \neq h_{22}$, $\boldsymbol{A}_n\boldsymbol{B}$ has full column rank. This means that when $\delta$ is small enough (from Example 3.1 we know $\tilde{\delta} = \sqrt{2}|h_{11} - h_{22}|$), the model rSpecT is infeasible.

## D   Proofs of Properties of (r)LogSpecT

### D.1   Proof of Proposition 3.4

Since the constraint set $\mathcal{S}$ is a cone, it follows that for all $\gamma > 0$, $\gamma\mathcal{S} = \mathcal{S}$. Then, we know that

$$\begin{aligned} \mathrm{Opt}(\boldsymbol{C}, \alpha) &= \operatorname*{argmin}_{\boldsymbol{S}\in\mathcal{S}, \boldsymbol{C}\boldsymbol{S}=\boldsymbol{S}\boldsymbol{C}} \|\boldsymbol{S}\|_{1,1} - \alpha\mathbf{1}^\top \log(\boldsymbol{S}\mathbf{1}) \\ &= \gamma \cdot \operatorname*{argmin}_{\gamma\boldsymbol{S}\in\mathcal{S}, \boldsymbol{C}\gamma\boldsymbol{S}=\gamma\boldsymbol{S}\boldsymbol{C}} \|\gamma\boldsymbol{S}\|_{1,1} - \alpha\mathbf{1}^\top \log(\gamma\boldsymbol{S}\mathbf{1}) \\ &= \gamma \cdot \operatorname*{argmin}_{\boldsymbol{S}\in\frac{1}{\gamma}\mathcal{S}, \boldsymbol{C}\boldsymbol{S}=\boldsymbol{S}\boldsymbol{C}} \gamma\|\boldsymbol{S}\|_{1,1} - \alpha\mathbf{1}^\top \log(\boldsymbol{S}\mathbf{1}) \\ &= \gamma \cdot \operatorname*{argmin}_{\boldsymbol{S}\in\mathcal{S}, \boldsymbol{C}\boldsymbol{S}=\boldsymbol{S}\boldsymbol{C}} \|\boldsymbol{S}\|_{1,1} - \frac{\alpha}{\gamma}\mathbf{1}^\top \log(\boldsymbol{S}\mathbf{1}) \\ &= \gamma\,\mathrm{Opt}(\boldsymbol{C}, \alpha/\gamma), \end{aligned}$$

where the third equality is from the basic calculus rule of the logarithm function. Set $\gamma = \alpha$ and then $\mathrm{Opt}(\boldsymbol{C}, \alpha) = \alpha\,\mathrm{Opt}(\boldsymbol{C}, 1)$, which completes the proof.

### D.2   Proof of Proposition 3.6

The proof will be conducted by constructing a feasible solution for rLogSpecT. Recall that $\boldsymbol{A}_n = \boldsymbol{I} \otimes \boldsymbol{C}_n - \boldsymbol{C}_n \otimes \boldsymbol{I}$ and the matrix $\boldsymbol{B} \in \mathbb{R}^{m^2 \times m(m-1)/2}$ that maps a non-negative vector to the vectorization of a valid adjacency matrix. Let $\boldsymbol{S} = \min\{\frac{\delta}{\|\boldsymbol{A}_n\boldsymbol{B}\boldsymbol{s}\|_2}, 1\} \cdot \mathrm{mat}(\boldsymbol{B}\boldsymbol{s})$ with $\boldsymbol{s} \in \mathbb{R}^{(m-1)m/2}$ being a non-negative vector, where $\mathrm{mat}(\cdot)$ is the matricization operator. Note that

$$\mathrm{vec}(\boldsymbol{C}_n\boldsymbol{S} - \boldsymbol{S}\boldsymbol{C}_n) = (\boldsymbol{I} \otimes \boldsymbol{C}_n - \boldsymbol{C}_n \otimes \boldsymbol{I})\,\mathrm{vec}(\boldsymbol{S}) = \boldsymbol{A}_n\mathrm{vec}(\boldsymbol{S}).$$

Then, we know that

$$\|\boldsymbol{C}_n\boldsymbol{S} - \boldsymbol{S}\boldsymbol{C}_n\|_F = \|\mathrm{vec}(\boldsymbol{C}_n\boldsymbol{S} - \boldsymbol{S}\boldsymbol{C}_n)\|_2 = \min\left\{\frac{\delta}{\|\boldsymbol{A}_n\boldsymbol{B}\boldsymbol{s}\|_2}, 1\right\} \cdot \|\boldsymbol{A}_n\boldsymbol{B}\boldsymbol{s}\|_2 \leq \delta.$$