# OpenReview forum: "LogSpecT: Feasible Graph Learning Model from Stationary Signals with Recovery Guarantees"
_NeurIPS.cc/2023/Conference — NeurIPS 2023 poster_

### Official Review · Reviewer_U8JP · 2023-07-07

**Soundness:** 3 good
**Presentation:** 3 good
**Contribution:** 3 good
**Rating:** 6
**Confidence:** 4

**Summary:**

This paper studies the problem of graph learning from stationary signals. The authors propose an algorithmic framework, LogSpecT, which addresses the shortcomings of an existing body of work that relies on graph learning using spectral templates.



**Strengths:**

1. The problem setting is graph learning for stationary graphs using spectral templates. This is a problem of interest in the graph signal processing domain and is motivated by practical applications. The major technical contribution of the paper is to solve the problem by introducing log barrier in the objective function. Introduction of log barrier facilitates tackling the graph learning scenarios that were not feasible by existing algorithms in this domain.
2. The paper is well written and the technical content is easy to follow.
3. The experiments demonstrate the LogSpecT ourtperforms the previous versions of spectral templates and two other baselines from the literature in the task of graph learning. Moreover, it is demonstrated that previous versions of graph learning algorithms may encounter infeasibility scenarios on real datasets with high likelihood.

**Weaknesses:**

There may be a minor concern that the technical contributions of the paper are narrowly focused on overcoming the shortcomings of graph learning with spectral templates.

**Questions:**

Could you comment on the computational complexity of LogSpecT? In particular, I am curious whether the operations involving $C_n$ in Section 5 could be computationally prohibitive for high dimensional datasets.

**Limitations:**

Limitations have been adequately discussed.

---

> ### Author Rebuttal · Authors · 2023-08-09
>
> Thank you for your comments. We now provide responses to the questions and concerns you have raised.
>
> **Q1:** There may be a minor concern that the technical contributions of the paper are narrowly focused on overcoming the shortcomings of graph learning with spectral templates.
>
> **Answer:** It appears to be a strict requirement that the graph signals are stationary. This assumption starts to gain popularity as the extension of the stationarity defined on the regular domain to the graph data [G15]. Since then, many GSP techniques have been developed under this assumption, e.g. graph learning under stationarity [SM$^+$], low-passness detection under stationarity [ZHW23] and multi-attribute graph signal processing under stationarity [ZHW22]. Along these works, many real datasets have been verified to be stationary or nearly stationary. For example, [G15] showed that a part of the weather data can be explained by stationary graph signals and [PV17] showed that the well-known USPS dataset and the CMUPIE set of cropped faces are nearly stationary. In a nutshell, the stationarity assumption has been a cornerstone in the GSP community. Moreover, it is quite insightful and interesting to push beyond the stationarity assumption and tackle the case when the graph signals are non-stationary. This is one of our future directions.
>
> **Q2:** Could you comment on the computational complexity of LogSpecT? In particular, I am curious whether the operations involved in Section 5 could be computationally prohibitive for high dimensional datasets.
>
> **Answer:** We consider the complexity in each iteration roughly when the number of nodes $m$ is larger than the number of observations $n$. In the update of $Z$ and $q$, the complexity is decided by the the calculation of $C_nS^{(k)}$. This can be written as $(1/nX)(X^\top S^{(k)})$, where $X \in \mathbb{R}^{m\times n}$ is the data matrix. Hence, the complexity of this step is in the order of $nm^2 + nm$. In the update of $S$, the complexity is decided by the calculation of $C^{(k)}$. The calculation of $C_n\Lambda$ can be conducted similarly, yielding the complexity of $nm^2 + nm$. In the calculation of $S^{(k)}C_n^2$, we may decompose it as $\frac{1}{n^2}(S^{(k)}X)(X^\top X)(X^\top)$. The complexity in this step is in the order of $nm^2+n^2m+nm$. In the calculation of $C_nS^{(k)}C_n$, we may calculate it as $\frac{1}{n^2}X(X^\top S^{(k)}X)X^\top$. The calculation involved here is in the order of $nm^2 + mn^2$. The update of the dual variables can be decomposed as the summation of the components calculated in the update of the primal variables. In summary, the total complexity of each iteration is in the order of $nm^2 + n^2m + nm$. The coefficients can be improved by storing some common components, e.g. $S^{(k)}X$ and so on. This would be quite efficient even in a high dimensional dataset.
>
> [G15] Girault, B. Stationary graph signals using an isometric graph translation. In Proceedings of the
> 23rd European Signal Processing Conference (EUSIPCO 2015), pages 1516–1520. IEEE, 2015.
>
> [SM$^+$17] Segarra, S., et al. Network topology inference from spectral templates. IEEE Transactions on Signal and Information Processing over Networks, 3(3):467–483, 2017.
>
> [ZHW22] Zhang, C., et al. Product graph learning from multi-attribute graph signals with inter-layer coupling. arXiv preprint arXiv:2211.00909, 2022.
>
> [ZHW23] Zhang, C., et al. Detecting low pass graph signals via spectral pattern: Sampling complexity and applications. arXiv preprint arXiv:2306.01553, 2023.
>
> [PV17] Perraudin, N and Vandergheynst, P. Stationary signal processing on graphs. IEEE Transactions on Signal Processing, 65(13):3462–3477, 2017.

---

> > ### Comment · Reviewer_U8JP · 2023-08-18
> >
> > Thank you for providing a detailed response.

---

> > > ### Author Response · Authors · 2023-08-19
> > >
> > > Thank you for your time.

---

> ### Comment · Area_Chair_5ADJ · 2023-08-18
>
> Hi,
>
> Could you please acknowledge (at least) the authors' rebuttal, and engage in the discussion if you still have concerns?
>
> Thanks in advance for helping NeurIPS reviewing process,
>
> Best,
> AC

---

### Official Review · Reviewer_f5Sh · 2023-07-07

**Soundness:** 4 excellent
**Presentation:** 4 excellent
**Contribution:** 4 excellent
**Rating:** 8
**Confidence:** 4

**Summary:**

In this paper, the authors considered the problem of learning graphs from stationary signals. In order to overcome the infeasibility issue of an existing method called rSpecT, the authors proposed a novel formulation by introducing the log barrier term to learn graphs without isolated nodes. The feasibility can be guaranteed by the new formulation. Furthermore, the recovery guarantees of the new formulation was established, and an efficient algorithm based on linearized ADMM was proposed to solve the formulation.

**Strengths:**

This paper is very interesting, with novel ideas to solve an issue of a well-known method in graph signal processing. The authors proposed a novel formulation to overcome the infeasibility issue of rSpecT and established the theoretical recovery guarantees. Furthermore, an efficient algorithm based on linearized ADMM was designed to solve the formulation. Numerical results demonstrated the stability and superiority of the new method.


**Weaknesses:**

1. The descriptions of the experiments conducted on real networks require further improvement. Specifically, please provide more details regarding the number of observations, the nature of these observations, and what the edges represent within the networks.

2. As observed in Figures 2 and 4, the F-measure does not increase to 1, even when given a sufficiently large number of observations. The authors should provide commentary on this phenomenon to address any potential concerns.

**Questions:**

Please answer questions from Weakness.

**Limitations:**

Yes

---

> ### Author Rebuttal · Authors · 2023-08-09
>
> Thank you so much for your positive comments and advice. We hope the following clarification can answer your questions.
>
> **Q1:** The descriptions of the experiments conducted on real networks require further improvement. Specifically, please provide more details regarding the number of observations, the nature of these observations, and what the edges represent within the networks.
>
> **Answer:** The experiments on real networks are conducted on graphs from the Protein database [BOS$^+$05] and the Reddit database [YV15]. The Protein database is a Bioinformatics dataset, where nodes are secondary structure elements (SSEs) and there is an edge between two nodes if they are neighbors in the amino-acid sequence or in 3D space. The Reddit database is a social network dataset. In this dataset, each graph corresponds to an online discussion thread, where nodes correspond to users, and there is an edge between two nodes if at least one of them responded to another’s comments. The observed graph signals are then generated synthetically according to the generative model (1), with the graph filtering chosen as $H(S) = \exp{\frac{1}{2}S}$. This is to make sure that the observed graph signals are stationary. For the experiments of real signals, we refer the reviewer to our global response, where we discuss the experiments on the handwritten digits USPS dataset, which is verified to be nearly stationary in [PV17].
>
> **Q2:** As observed in Figures 2 and 4, the F-measure does not increase to 1, even when given a sufficiently large number of observations. The authors should provide commentary on this phenomenon to address any potential concerns.
>
> **Answer:** Figure 2 presents the results of rLogSpecT on BA graphs. As rLogSpecT is an approximation of LogSpecT, its performance will approach that of LogSpecT when the sample size is growing. However, on BA graphs, even LogSpecT cannot achieve 1 F-measure. Hence, the approximation model rLogSpecT cannot achieve 1 F-measure, no matter how large the sample size is. The phenomenon observed in Figure 4 is similar. The ideal model cannot achieve perfect recovery on any real networks, let alone the approximation model rLogSpecT. It remains an open and intriguing problem in GSP how broad the class of networks is, on which LogSpecT can achieve perfect recovery. This is also one of our future directions.
>
> [BOS$^+$05] Borgwardt, K., et al. Protein function prediction via graph kernels. Bioinformatics, 21:47–56, 2005.
>
> [YV15] Yanardag, P and Vishwanathan, S. Deep graph kernels. In Proceedings of the 21th ACM SIGKDD international conference on knowledge discovery and data mining, pages 1365–1374, 2015.
>
> [PV17] Perraudin, N and Vandergheynst, P. Stationary signal processing on graphs. IEEE Transactions on Signal Processing, 65(13):3462–3477, 2017.

---

> > ### Comment · Reviewer_f5Sh · 2023-08-13
> >
> > Thank you for your insightful and thorough response. Your detailed explanations have not only effectively addressed the concerns that were raised, but have also provided valuable clarity on the key points of the study.

---

> > > ### Author Response · Authors · 2023-08-14
> > >
> > > Thank you for your time.

---

### Official Review · Reviewer_xFRE · 2023-07-08

**Soundness:** 3 good
**Presentation:** 4 excellent
**Contribution:** 3 good
**Rating:** 7
**Confidence:** 3

**Summary:**

This paper addresses the problem of learning a graph from stationary graph signals. To this end, the authors introduce two new optimisation problems called LogSpecT and rLogSpecT.  They prove some recovery guarantees on the optimal value of (r)LogSpecT and give a convergence rate when the signals are sub-Gaussian. They also provide an efficient solver based on L-ADMM that admits closed solutions at each substep.  Finally, some empirical results highlight the performance of their methods.

**Strengths:**

1\ The paper is very well written and easy to follow

2\ The problem is well-motivated and the solution seems theoretically sound

3\ The paper is solid and almost all aspects encountered in the graph learning problem are covered (optimization, recovery guarantees, convergence,...)

**Weaknesses:**

1\ There are no error bars. They should be added to really show the results of each method and to be able to compare them.

2\ A table with all the measurements reported could be a good thing. For example, in all the experiments, only the F measure is given, but precision and recall are also important for assessing the quality of the learned graphs.

3\ An illustration comparing the real graph with the learned graph (for example by plotting the adjacency matrix) could be interesting.

4\ The complexity and the time until convergence are not given for the methods. An additional experiment on this aspect should be added.

=======
Remark

the step size tau is never defined

**Questions:**

1\ What is the intuition behind the need for a delta greater than a certain value in Theorem 4.1?

2\ In the GSP literature, there are many more recent methods than the one of Kalofolias for learning the graph. Why choose to compare your methods only to the one of Kalofolias?

3\ Furthermore, why exclude this method from the comparison in the results of the section 6.3?

4\ Experiments are made with a maximum of 60 nodes. What is the maximum number of nodes that can be learned in a reasonable time? if possible, can we have some examples?

=====

More open questions:

5\ Is it possible to link the optimization problem to a maximum likelihood estimate?

6\ It seems that the filter h only has an impact on the covariance estimate. Is it true? If the answer is "no", would it be a good idea (if possible) to include information about the h filter in the optimization problem?

**Limitations:**

-

---

> ### Author Rebuttal · Authors · 2023-08-09
>
> Thanks for your comments and advice. We hope that the clarification may help to clear your concerns.
>
> **Q1:** - There are no error bars. They should be added to really show the results of each method and to be able to compare them. - A table with all the measurements reported could be a good thing. For example, in all the experiments, only the F measure is given, but precision and recall are also important for assessing the quality of the learned graphs. - An illustration comparing the real graph with the learned graph (for example by plotting the adjacency matrix) could be interesting.
>
> **Answer:** Thank you for your helpful advice on presenting the results. We showed the error bars in the real data experiments for fair comparison. However, we omit the error bars in the synthetic data experiments as both models achieve nearly perfect recovery or the difference between them is too obvious. We would like to present all the measurements in the appendix, together with an illustration comparing the groudtruth graph with the learned graph.
>
> **Q2:** - The complexity and the time until convergence are not given for the methods. An additional experiment on this aspect should be added. - The step size tau is never defined.
>
> **Answer:** For the running time and complexity of our proposed L-ADMM method, we refer the reviewer to our global response and our response to Reviewer U8JP, where we discussed the complexity in detail and added an experiment to present the residuals until convergence. Also, thank you for your kind reminder.  $\tau$ in the paper is defined as the parameter in the linearization process and can be viewed as a step size in updating $S$. We will add these clarifications to our paper.
>
> **Q3:** What is the intuition behind the need for a delta greater than a certain value in Theorem 4.1?
>
> **Answer:** The intuition that $\delta$ should be larger than a certain value is that $\delta$ is to reflect our view on the error of estimating the covariance matrix. We can bear with the overestimate of the errors while the underestimate is unacceptable. From a theoretically viewpoint, the $\delta$ should be set to ensure that the feasible set of rLogSpecT at least contains the optimal solutions to LogSpecT. This part is required in our proof. Although this condition appears to be stringent, our analysis in Corollary 4.8 then shows that if $\delta$ is set to decrease in a proper rate, the condition will hold with high probability.
>
> **Q4:** In the GSP literature, there are many more recent methods than the one of Kalofolias for learning the graph. Why choose to compare your methods only to the one of Kalofolias?
>
> **Answer:** We only compared Kalofolias' method in the experiments and omitted other models for the reason that Kalofolias' method is the only graph learning model with the log barrier. Since our model is another graph learning model with the log barrier, it is natural to compare our model with Kalofolias'. We didn't compare our model with the more recent models as they are based on the probabilistic graphical model instead of the stationarity assumption. However, for fair comparison, we present the experiment results of ALPE, NGL-SCAD, NGL-MCP on the Protein and Reddit datasets in the global response.
>
> **Q5:** Furthermore, why exclude this method from the comparison in the results of the section 6.3?
>
> **Answer:** We excluded the performance of Kalofolias' method in section 6.3 for the reason that SpecT is the only efficient model for graph learning from stationary signals and Kalofolias' method is designed for smooth graph signals instead of stationary graph signals. However, for fair comparison, we add the experiments of Kalofolias' method on 50-node ER graphs with different sparsity levels in the global response. The results show that this model cannot learn graphs from the stationary signals.
>
> **Q6:** Experiments are made with a maximum of 60 nodes. What is the maximum number of nodes that can be learned in a reasonable time? if possible, can we have some examples?
>
> **Answer:** The experiment results on 100 node graphs are shown in the global response, which will take around 20 seconds to learn a graph. We note that the capability of our proposed models and algorithms depends not only on the devices, but also on the way to implement the algorithms. As L-ADMM is the first algorithm designed for our proposed LogSpecT, we will investigate the more efficient algorithms based on it in the future.
>
> **Q7:** Is it possible to link the optimization problem to a maximum likelihood estimate? It seems that the filter h only has an impact on the covariance estimate. Is it true? If the answer is "no", would it be a good idea (if possible) to include information about the $h$ filter in the optimization problem?
>
> **Answer:** The two additional open questions are quite insightful and interesting. For the first one, it is a well-known open problem in GSP. To understand why LogSpecT performs well is essential for designing more efficient models for graph learning from stationary signals and further pushing beyond the stationarity assumption. It is actually one of our future directions. For the second question, the impact of $h$ may lie in two folds. On one hand, it affects the estimated covariance matrix $C_n$, which may consequently affect the efficiency of our proposed algorithm. On the other hand, the prior knowledge of $h$ may help to enlarge the class of graphs that can be perfectly recovered by the spectral template. To understand the influence of $h$ in a rigorous way deserves further investigation.

---

> > ### Comment · Reviewer_xFRE · 2023-08-14
> >
> > Thank you for your clear and thorough response.

---

> > > ### Author Response · Authors · 2023-08-15
> > >
> > > Thank you for your appreciation.

---

### Official Review · Reviewer_xuW7 · 2023-07-10

**Soundness:** 3 good
**Presentation:** 4 excellent
**Contribution:** 3 good
**Rating:** 6
**Confidence:** 1

**Summary:**

In this paper, the authors improve upon an existing method, rSpecT, to impute graphs from stationary signals. The latter frames the recovery of the (weighted) adjacency matrix as a convex optimization problem with constraints —- namely the commutativity of the covariance matrix and the adjacency matrix, as well as the first row sum to be equal to 1, in order to avoid the trivial 0 solution. Here, the authors show that this yields infeasible solutions in a number of cases. Consequently, the authors propose to replace the row sum constraint by a log barrier. They show that the solution is always feasible (however, it ceases to be unique).  They then show through a set of experiments that their solution outperforms the state of the art approaches in recovering the underlying graph.

**Strengths:**

First, a disclaimer: although familiar with GSP, I am not an expert in the graph recovery literature. My comments must thus be taken sparingly.

Overall, the paper is clearly written and motivated.
The discussion on the infeasibility of the current state of the art method, rSpecT, was interesting. The solution that the authors bring is therefore well motivated.

**Weaknesses:**

As highlighted before, I am not an expert in the field and I am not able to put this paper in context of the general literature on the topic.

A few questions however:
- the authors claim that their method is more computationally efficient with the use of ADMM. Do they have plots showing the running time? ADMM, although fast at each iteration, can be slow to converge.
- the experiments could have been expanded: in particular, the authors only present ER and BA graphs, but they could have varied the parameters to try and assess whether their method is robust across all sorts of topologies.


**Questions:**

See weaknesses.

**Limitations:**

The method could have been more extensively evaluated.

---

> ### Author Rebuttal · Authors · 2023-08-09
>
> Thank you for your comments and advice. We hope the clarification can clear your concerns.
>
> **Q1:** The authors claim that their method is more computationally efficient with the use of ADMM. Do they have plots showing the running time? ADMM, although fast at each iteration, can be slow to converge.
>
> **Answer:** For the running efficiency of our proposed L-ADMM, we refer the reviewer to our global response and our response to Reviewer U8JP that talks about the complexity in each iteration. In the experiment, we show that L-ADMM can reach the accuracy of $10^{-6}$ in 3500 iterations for a graph with 100 nodes. Also, we can observe the local linear convergence for L-ADMM in our model, which indicates that our algorithm can run efficiently. We will add the detailed discussion in our paper.
>
> **Q2:** The experiments could have been expanded: in particular, the authors only present ER and BA graphs, but they could have varied the parameters to try and assess whether their method is robust across all sorts of topologies.
>
> **Answer:** In order to show the robustness LogSpecT, we investigate its performance when the sparsity level of the underlying graphs changes. Specifically, we set the parameter in the generation of ER graphs from 0.1 to 0.5, which represents the probability that an edge exists between two nodes. The number of nodes is set as 50 and the graph filter is chosen as QUA: $H(S) = S^2 + S + I$. The average results of 10 independent replicates are shown in Figure 1 in the global response. This figure demonstrates the robustness of LogSpecT.

---

> ### Comment · Area_Chair_5ADJ · 2023-08-18
>
> Hi,
>
> Could you please acknowledge (at least) the authors' rebuttal, and engage in the discussion if you still have concerns?
>
> Thanks in advance for helping NeurIPS reviewing process,
>
> Best,
> AC

---

### Official Review · Reviewer_SR4h · 2023-07-12

**Soundness:** 2 fair
**Presentation:** 2 fair
**Contribution:** 2 fair
**Rating:** 3
**Confidence:** 1

**Summary:**

This is an emergency review, regrettably, the paper is outside of my expertise.

Review:
This is a theoretical work concerned with graph learning from signals. The work identifies an issue with the feasibility of a previous well known model in the field, SpecT, then introduces a novel model, LogSpecT, for which this issue does not occur.

While I regrettably could not comprehend the main content, the topic, general presentation, and references make me believe that this paper is topically not suited for neurIPS. I concede that the CfP of neurIPS is somewhat vague and broad, but based on the common topics in the field, I believe for the conference this is very much a niche topic to which the vast majority of attendees would not be able to connect.

Superficial observations
 - There isn't any mention of the word "neural"
 - From 45 references, the vast majority is in signal processing. A single one is at neurIPS
 - The intro is exceedingly dry to read, with no motivating example, nor any illustration. Especially, what are examples of signals considered for graph learning? What would the stationarity assumption mean in these examples?

I understand that reading a superficial review like this may be frustrating to the authors, but can do no better. I would much suggest the authors consider a specialist forum, where this type of content might be better understood and appreciated.

**Strengths:**

see above

**Weaknesses:**

see above

**Questions:**

see above

**Limitations:**

see above

---

> ### Author Rebuttal · Authors · 2023-08-09
>
> Thanks for your comments. Here are some illustrations based on your questions/suggestions.
>
> **Q1:** There isn't any mention of the word "neural"
>
> **Answer:** Graph learning from signals studied in this paper has much relevance in the machine learning and neural science community.
> For example, it has broad applications in bioinformatics [RX$^+$21].
> More specifically, we consider the task of graph topology inference from observed data. It has been applied to recognize the functional connectivity related to neural activity patterns. The vast references in this field include [DE06, EO09, GX$^+$21], to name just a few. Due to limits of page length, we are only able to focus on the modeling issue and its theoretical guarantees in this paper.
>
> **Q2:** From 45 references, the vast majority is in signal processing. A single one is at NeurIPS
>
> **Answer:** We are working on a type of data that was newly proposed in the signal processing community and it has not gained much attention from the machine learning community. However, the task we were caring about has long been a focus in machine learning. For example, [K16, KP18] studied how to learn a large graph from smooth graph signals, [YCP20, YCP21, VYP22] investigated this task from the probabilistic graphical model.
>
> **Q3:** The intro is exceedingly dry to read, with no motivating example, nor any illustration. Especially, what are examples of signals considered for graph learning? What would the stationarity assumption mean in these examples?
>
> **Answer:** Thank you for your comments in the introduction part. In practice, the meaning of signals and graphs depends on the scenarios and the models. The signals can be understood as the nodal features in the graph learning task. For instance, in the financial market, the companies can be modeled as the nodes and the signals are the stock returns of each company. The graph learning task is then to investigate the relationship between the companies' stock returns. The relevance of the stationary assumption in several real data sets has been revealed in several existing literature.  [G15] showed that a part of the weather data can be explained by stationary graph signals. [PV17] showed that the well-known USPS dataset and the CMUPIE set of cropped faces are close to stationary. [SM$^+$17] compared the performance of learning protein structures with/without the stationary assumption and validated this assumption. [ZHW22] applied the stationary assumption to US Senate Roll Calls and achieved good performance.  In a nutshell, the stationarity assumption is a theoretical property that stems from the stationary time series and has been verified in several real data. This property has to be understood more from a theoretical viewpoint instead of the intuition.
>
>
> [RX$^+$21] Rui, L., et al. Graph signal processing, graph neural network and graph learning on biological data: a systematic review. IEEE Reviews in Biomedical Engineering, 2021.
>
> [DE06] Danielle, B. and ED, B. Small-world brain networks. The neuroscientist, 12(6):512–523, 2006.
>
> [EO09] ED, B. and Olaf, S. Complex brain networks: graph theoretical analysis of structural and functional systems. Nature reviews neuroscience, 10(3):186–198, 2009.
>
> [GX$^+$21] Gao, S., et al. Smooth graph learning for functional connectivity estimation. NeuroImage, 239:118289, 2021.
>
> [K16] Kalofolias, V. How to learn a graph from smooth signals. In Proceedings of the 19th International Conference on Artificial Intelligence and Statistics (AISTATS 2016), pages 920–929. PMLR, 2016.
>
> [KP18] Kalofolias, V. and Perraudin, N. Large scale graph learning from smooth signals. In International Conference on Learning Representations, 2018.
>
> [YCP20] Ying, J., et al. Nonconvex sparse graph learning under Laplacian constrained graphical model. Advances in Neural Information Processing Systems, 33:7101–7113, 2020.
>
> [YCP21] Ying, J., et al. Minimax estimation of Laplacian constrained precision matrices. In International Conference on Artificial Intelligence and Statistics, pages 3736–3744. PMLR, 2021.
>
> [VYP22] Vinícius, J., et al. Learning bipartite graphs: Heavy tails and multiple components. Advances in Neural Information Processing Systems, 35:14044–14057, 2022.
>
> [G15] Girault, B. Stationary graph signals using an isometric graph translation. In Proceedings of the 23rd European Signal Processing Conference (EUSIPCO 2015), pages 1516–1520. IEEE, 2015.
>
> [PV17] Perraudin, N and Vandergheynst, P. Stationary signal processing on graphs. IEEE Transactions on Signal Processing, 65(13):3462–3477, 2017.
>
> [SM$^+$17] Segarra, S., et al. Network topology inference from spectral templates. IEEE Transactions on Signal and Information Processing over Networks, 3(3):467–483, 2017.
>
> [ZHW22] Zhang, C., et al. Product graph learning from multi-attribute graph signals with inter-layer coupling. arXiv preprint arXiv:2211.00909, 2022.

---

> > ### Comment · Reviewer_SR4h · 2023-08-14
> >
> > Thank you for the comprehensive response.

---

> > > ### Author Response · Authors · 2023-08-14
> > >
> > > You are welcome. Hope our response can clear up all your concerns.

---

### Author Rebuttal · Authors · 2023-08-09

We thank all the reviewers for your insightful comments and helpful advice. In the global response, we explain the additional experiments and results shown in the attached pdf. We hope our clarifications can help to clear up reviewers' concerns.

**1. Stability of LogSpecT and comparison with Kalofolias' method.**

We change the probability that an edge exists in the generation of 50-node ER graphs and use the QUA graph filter to generate stationary graph signals. LogSpecT and Kalofolias' method are conducted on the signals. We present the average F-measure over 10 independent replicates in Figure 1. Specifically, the parameter in Kalofolias' method is chosen from $\{0.01, 0.1, 1, 10, 100, 1000\} $ that yields the highest F-measure. The results show that LogSpecT performs well and stably regardless of the sparsity level. However, Kalofolias' method fails on the stationary graph signals.

**2. Comparison with more recent models on real networks and synthetic signals.**

We test ALPE [YCP21], NGL-MCP, and NGL-SCAD [YCP20] on the Protein and Reddit datasets with the synthetic stationary graph signals generated from lowpass-EXP: $H(S) = \exp{S}$. The true covariance matrix is assumed to be available. The mean and standard deviation of the F-measure from these models are collected in Table 1. The results indicate that the tested advanced algorithms based on the probabilistic graphical model perform poorly on these two datasets with stationary graph signals. A possible reason is that the stationary assumption does not fit the probabilistic graphical model.

**3. Efficiency of the L-ADMM algorithm.**

Since there does not literally exist customized algorithm for rLogSpecT, we compare our proposed L-ADMM with CVXPY [SS16]. The solver used in CVXPY is MOSEK. We conduct the experiments on 100-node BA graphs with 10000 low-pass EXP generated stationary graph signals. $\delta$ is set as $10\sqrt{\frac{\log n}{n}}$. For L-ADMM algorithm, the accuracy is set as $10^{-6}$ and the initialization is set as zero. The running time for L-ADMM takes around $20$ seconds while the solver takes over $110$ seconds. Note that the complexity of each iteration in L-ADMM can be further improved with the treatment of matrix multiplication mentioned in our response to Reviewer U8JP. We present the primal residual and the dual residual with respect to the iteration index in Figure 2. The results corroborate the efficiency of our proposed algorithm. Furthermore, the local linear convergence can be observed from the residuals, which can lead to fast convergence to high accuracy. It is one of our future directions to prove this property theoretically.

**4. Experiments on real datasets: the handwritten digits USPS dataset.**

As shown in [PV17], USPS dataset is nearly stationary with respect to the nearest neighbour graph. This dataset collects images of handwritten digits. In the experiment, each pixel is viewed as a node and the value on the pixel is view as the graph signal. We follow the approach in [PV17] to construct the 20 nearest neighbour graph, on which the data is verified to be nearly stationary. More specifically, we pick the 1296 digit 1 images. The weights between two nodes are decided by the Gaussian radial basis function. It turns out that the stationarity measure $s := \Vert\operatorname{diag}(U^\top C_n U)\Vert_2 / \Vert U^\top C_n U\Vert_F$ equals 0.78 in this dataset. Here $U$ is the eigen-matrix of the constructed graph and $C_n$ is the covariance matrix of the observed graph signals. We view the 20 nearest neighbour graph as the groundtruth and compare the learned graph from rLogSpecT with it. We present the subgraphs consisting of the top 10 nodes with the largest numbers of neighbours in the two graphs respectively in Figure 3. The F-measure of these two subgraphs is 0.96. This result corroborates the efficiency of our proposed rLogSpecT model.

[PV17] Perraudin, N and Vandergheynst, P. Stationary signal processing on graphs. IEEE Transactions on Signal Processing, 65(13):3462–3477, 2017.

[YCP20] Ying, J., et al. Nonconvex sparse graph learning under Laplacian constrained graphical model. Advances in Neural Information Processing Systems, 33:7101–7113, 2020.

[YCP21] Ying, J., et al. Minimax estimation of Laplacian constrained precision matrices. In International Conference on Artificial Intelligence and Statistics, pages 3736–3744. PMLR, 2021.

[SS16] Steven, D. and Stephen, B. Cvxpy: A python-embedded modeling language for convex optimization. The Journal of Machine Learning Research, 17(1):2909–2913, 2016.

---

### Comment · Area_Chair_5ADJ · 2023-08-13

Thanks to all reviewers and authors for their work on this submission.

As the discussion period starts, I want to make sure that reviewers have read the author's response.

This can be done either by communicating with authors, or in private conversation within the reviewing team.

---

### Decision · Program_Chairs · 2023-09-21

**Decision:**

Accept (poster)

**Comment:**

This paper is concerned with learning graph structures from signals with recovery guarantees. It was overall well received by the reviewers, and the rebuttal allowed to fix minor concerns.
Please incorporate the remaining reviewers' feedback for the camera-ready version.